



# The dual-field-of-view polarization lidar technique: A new concept in monitoring aerosol effects in liquid-water clouds — Case studies

Cristofer Jimenez[1], Albert Ansmann[1], Ronny Engelmann[1], David Donovan[2], Aleksey Malinka[3], Patric Seifert[1], Martin Radenz[1], Zhenping Yin[1,4,5], Robert Wiesen[1], Johannes Bühl[1], Jörg Schmidt[6], Ulla Wandinger[1], and Boris Barja[7]

[1]Leibniz Institute for Tropospheric Research, Leipzig, Germany
[2]Royal Netherlands Meteorological Institute (KNMI), De Bilt, The Netherlands
[3]National Academy of Sciences of Belarus, Minsk, Belarus
[4]School of Electronic Information, Wuhan University, Wuhan, China
[5]Key Laboratory of Geospace Environment and Geodesy, Ministry of Education, Wuhan, China
[6]Institute of Meteorology, University of Leipzig, Leipzig, Germany
[7]Atmospheric Research Laboratory, University of Magallanes, Punta Arenas, Chile

**Correspondence:** Cristofer Jimenez (jimenez@tropos.de)

**Abstract.**

In a companion article (Jimenez et al., 2020), we introduced a new lidar method to derive microphysical properties of liquid-water clouds (cloud extinction coefficient, droplet effective radius, liquid-water content, cloud droplet number concentration $N_\mathrm{d}$) at a height of 50-100 m above cloud base together with aerosol information (aerosol extinction coefficients, cloud

5   condensation nucleus concentration $N_\mathrm{CCN}$) below the cloud layer so that detailed studies of the influence of given aerosol conditions on the evolution of liquid-water cloud layers with high temporal resolution solely based on lidar observations became possible now. The novel cloud retrieval technique makes use of lidar observations of the volume linear depolarization ratio at two different receiver field-of-views (FOVs). In this part 2, the new dual-FOV polarization lidar technique is applied to cloud measurements in pristine marine conditions at Punta Arenas in southern Chile. A multiwavelength polarization Raman

10   lidar, upgraded by integrating a second polarization-sensitive channel to permit depolarization ratio observations at two FOVs, was used for these measurements at the southernmost tip of South America. Two case studies are presented to demonstrate the potential of the new lidar technique. Successful aerosol-cloud-interaction (ACI) studies based on measurements with the upgraded aerosol-cloud lidar in combination with a Doppler lidar of the vertical wind component could be carried with one-minute temporal resolution at these pristine conditions. In a stratocumulus layer at the top of the convective boundary layer, we

15   found values of $N_\mathrm{d}$ and $N_\mathrm{CCN}$ (for 0.2% water supersaturation) ranging from 15-100 cm$^{-3}$ and 75-200 cm$^{-3}$, respectively, and associated activation ratios ($N_\mathrm{d}/N_\mathrm{CCN}$) of 0.25-0.5 during updraft periods. The studies of the aeorosol impact on cloud properties yielded ACI values close to 1. The impact of aerosol water-uptake on the ACI studies was analyzed with the result that the highest ACI values were obtained when considering aerosol proxies (light extinction coefficient $\alpha_\mathrm{par}$ or $N_\mathrm{CCN}$) measured at heights about 500 m below cloud base (and thus for dry aerosol conditions).

©c Author(s) 2020. CC BY 4.0 License.





*Copyright statement.* TEXT

# 1 Introduction

Numerous details and aspects of aerosol-cloud interaction (ACI) are not well understood and thus not well considered and parameterized in weather and climate models. The reason for this gap in our knowledge is closely linked to the lack of adequate

measurements, observational concepts, instrumentation, tools, and techniques for a detailed, continuous (camera-like) monitoring of cloud processes in a variety of aerosol environmental and meteorological conditions. Such a continuous monitoring is only possible in well designed ground-base remote sensing network structures. Network supersites, distributed around the world, preferably in hotspot regions of anthropogenic activities and climate change as well as in rural background regions, need to cover profiling of aerosol mixtures and their aerosol-type properties, cloud microphysical, optical, and cloud-type

(phase) properties, and meteorological parameters such as temperature, relative humidity, and wind, especially of the vertical-wind component and thus of updraft, downdraft, and wave characteristics. Aerosol influences on low level liquid-water clouds over mixed-phase clouds to tropopause cirrus need to be monitored. Measurements of cloud-relevant aerosol parameters must even include heights within the lower stratosphere which may serve as source for ice-nucleating particles of heterogeneous ice formation in high level cirrus layers. In Europe, ACTRIS (Aerosols, Clouds, and Trace gases Research InfraStructure,

https://www.actris.eu/) with its network structures Cloudnet (Illingworth et al., 2007) and EARLINET (European Aerosol Research Lidar Network) (Pappalardo et al., 2014) is responsible for the build up of the necessary aerosol-cloud monitoring infrastructure.

There is still a strong request for the development of new and robust aerosol and cloud profiling techniques (Grosvenor et al., 2018). As a contribution to improved ACI field studies with focus on liquid-water clouds, we developed a novel lidar

measurement concept that permits continuous, vertically resolved observations of cloud-relevant aerosol properties below cloud base, cloud microphysical properties in the lower part of the cloud layer, and of the vertical wind component below and within the cloud parcels with a temporal resolution of 30-120 s. The methodological framework is presented in part 1 (Jimenez et al., 2020). The selected measurement concept of combine aerosol lidar, cloud lidar, and wind Doppler lidar observations was already outlined and applied by Schmidt et al. (2014, 2015). However, a fast lidar technique for cloud observation (day

and night and with updraft-resolving temporal resolution) was introduced only recently (Jimenez et al., 2017, 2018). In part 1 (Jimenez et al., 2020), we presented the theoretical framework of the novel dual-FOV polarization lidar method which allows us to derive microphysical properties of liquid-water clouds such as droplet number concentration $N_\mathrm{d}$, effective radius $R_\mathrm{e}$ of the droplets, liquid water content $w_\mathrm{l}$ as well as the cloud extinction coefficient $\alpha$ in the cloud base region at 50 to 100 m above cloud base. Together with the recently developed method to derive height profiles of cloud condensations nucleus (CCN)

concentrations ($N_\mathrm{CCN}$) from aerosol extinction coefficients $\alpha_\mathrm{par}$, measured with the same polarization lidar (Mamouri and Ansmann, 2016) below cloud base, detailed studies of the impact of aerosol particles on the evolution of liquid water cloud layers became possible. The instrumental setup can be easily integrated in existing ground-based aerosol and cloud remote sensing network supersites as already demonstrated in the case of the mobile ACTRIS Cloudnet station LACROS (Leipzig





Aerosol and Cloud Observation System, http://lacros.rsd.tropos.de/) which is presently deployed for a long-term campaign in Punta Arenas, Chile, in the pristine marine environment of southernmost South America. The new dual-FOV polarization lidar technique can be easily implemented in wide-spread aerosol polarization lidars (of, e.g., EARLINET) with near-range and far-range receiver telecopes, as will be discussed below.

In this second article (part 2), we apply the new dual-FOV polarization lidar technique to recent aerosol and cloud observations at Punta Arenas, Chile, discuss the cloud retrieval uncertainties, compare the results with independent alternative cloud observations, and highlight the new potential of the lidar technique to significantly contribute to atmospheric and climate research in the field of ACI. In section 2, we briefly summarize the data analysis procedure to retrieve the aerosol and cloud parameters for in-depth ACI studies as extensively discussed and explained in part 1 (Jimenez et al., 2020). Section 3 provides

details of the integration of the dual-FOV polarization lidar technique into a Polly ($PO$ortab$Le$ $L$idar s$Y$stem) instrument (Engelmann et al., 2016; Baars et al., 2016). The upgraded Polly is part of the mobile Leipzig Aerosol and Clouds Remote Observation System LACROS consisting of the aerosol/cloud lidar, a Doppler lidar for profiling of the vertical wind component, and a cloud radar as the main profiling instruments. LACROS was continuously operated at Punta Arenas, Chile, in the framework of the long-term field campaign DACAPO-PESO (Dynamics, Aerosol, Cloud And Precipitation Observations in

the Pristine Environment of the Southern Ocean, https://dacapo.tropos.de) from November 2018 to the end of 2020. Details to the campaign and the goals of the investigations are outlined in Sect. 3.3. In the measurement section (Sect. 4), two case studies are presented. Case 1 (Sect.4.1) is shown to discuss the basic and principle features of the new cloud retrieval technique. The potential of the new dual-FOV lidar to contribute to ACI research is then illuminated in Sect. 4.2 (case study 2). Concluding remarks and an outlook are given in Sect. 5.

**2   Data processing scheme**

As outlined in part 1 (Jimenez et al., 2020), the basic motivation for the development of the dual-FOV polarization lidar technique was the need for simultaneous aerosol and cloud observations at day and nighttime with high temporal resolutions of the order of 30-120 s. The developed novel lidar method for liquid-water cloud observations is based on the measurement of the so-called volume linear depolarization ratio in the lower part of the water cloud at two different receiver FOV. The required

dual-FOV polarization lidar transmits linearly polarized laser pulses and detects the so-called cross- and co-polarized signal components. "Co" and "cross" denote the planes of polarization parallel and orthogonal to the plane of linear polarization of the transmitted laser pulses, respectively. The volume linear depolarization ratio is defined as the ratio of the cross- to the co-polarized signal and yields the basic information on the ratio of the cross-to-co-polarized backscatter coefficient. In water clouds, the depolarization ratio is sensitively influenced by multiple scattering and varies, e.g., with receiver FOV, cloud height,

and cloud droplet number concentration and size of the droplets as explained in detail in part 1. These relationships are used in the dual-FOV polarization lidar technique to retrieve the effective radius $R_e$ of the droplets and the cloud extinction coefficient $\alpha$ in the cloud base region at 50 to 100 m above cloud base by means of measured depolarization ratios at two FOVs, and, in the next step, to compute the liquid water content $w_l$ and the cloud droplet number concentration $N_d$ from the $R_e$ and $\alpha$ values.



Table 1 provides an overview of all steps of the comprehensive analysis of cloud and aerosol data obtained with the new dual-FOV polarization lidar developed in part 1 (Jimenez et al., 2020). The overall concept of lidar-based aerosol-cloud-interaction studies with focus on liquid-water clouds is illustrated in Fig. 4 in part 1. The dual-FOV polarization lidar technique allows us to derived simultaneously the microphysical properties of liquid-water clouds at a height of $z_{\mathrm{ref}}=z_{\mathrm{bot}}+75$ m above cloud base height $z_{\mathrm{bot}}$ and the aerosol proxies $\alpha_{\mathrm{par}}$ and $N_{\mathrm{CCN}}$ at height $z_{\mathrm{aer}}$ which can be freely selected and is typically about 250-750 m below cloud base to avoid aerosol water-uptake effects (Skupin et al., 2016; Haarig et al., 2017) on the ACI studies.

In this article (part 2), we apply the full methodology to a DACAPO-PESO measurement case collected at Punta Arenas on 22 March 2019 and explain all retrieval steps listed in Table 1. The second case, measured on 23 Februray 2019 is then presented to highlight the new potential of this novel lidar approach to significantly improve ACI studies in the case of liquid-water clouds. Before, we describe the lidar hardware needed for such observations and the way to obtain the required depolarization ratios from the measured lidar signals that serve as the basic input in the cloud data analysis scheme.

## 3    Instrument and experiment

### 3.1    Polly with dual-FOV capability

We implemented the dual-FOV polarization lidar technique in several lidars of the Leibniz Institute for Tropospheric Research (TROPOS) during the last years. The dual-FOV polarization lidar technique was firstly integrated into the EARLINET (European Aerosol Research Lidar Network) lidar MARTHA (Multiwavelength Atmospheric Raman lidar for Temperature, Humidity, and Aerosol profiling) (Jimenez et al., 2019). MARTHA was already equipped with the dual-FOV Raman lidar technique (Schmidt et al., 2013, 2014) so that direct comparisons of cloud observations with the Raman lidar and the polarization lidar method became possible. We found in general good agreement in the retrieval of cloud optical and microphysical properties (Jimenez et al., 2017, 2018). Encouraged by this successful comparisons, we stepped forward and equipped four Polly (*PO*ortab*L*e *L*idar s*Y*stem) instruments (Engelmann et al., 2016) with the new dual-FOV polarization lidar technique. These four lidar systems are and were involved in several long-term field activities at very different aerosol and environmental conditions, namely at Dushanbe, Tajikistan, in Central Asia (continuous measurements since June 2019), aboard the German ice breaker Polarstern (North Pole, a one-year campaign from October 2019 to September 2020), at Punta Arenas (about 2-year campaign from November 2018 to the end of 2020), and at Limassol, Cyprus (continuous measurements since July 2020). Aerosol retrieval methods and measurement examples can be found in Baars et al. (2016)and Hofer et al. (2017, 2020). Improved water vapor observations (water vapor mixing ration, relative humidiy) by combining lidar and regular photometer observations were recently discussed by Dai et al. (2018). In this section, we concentrate on the new approach of cloud measurements at two FOVs.

Figure 1 shows the transmitter and receiver configuration of the Polly instrument of the Punta Arenas remote sensing facility. The lidar is described in detail by Engelmann et al. (2016) and Hofer et al. (2017). Laser beam diameter and divergence are 45 mm and 0.2 mrad, respectively, after beam expansion. The polarization impurity (fraction of non linear polarized light) of the transmitted laser beam is less than 0.1%. The receiver unit consists of the near-range receiver part (red frame in Fig. 1),





optimized to provide particle backscatter and extinction profiles almost down to the ground determined from measured total and nitrogen Raman backscatter signals at 355, 387, 532, and 607 nm, and a far-range receiver part (blue frame). The diameter of the primary mirror of the far-range Newtonian telescope is 30cm. The overlap of the laser-beam with the receiver FOV is incomplete for heights below about 1 km above ground level (a.g.l.) and allows accurate aerosol and cloud profiling for heights

above about 500 m a.g.l., only (after the correction of the overlap effects). For the far-range channels, the selected $\mathrm{FOV_{in}}$ is 1.0 mrad. The FOV for the near-range channels is $\mathrm{FOV_{out}} = 2.0$ mrad. As can be seen in contrast to a classical polarization lidar as described above, the Polly instrument measures the cross-polarized and the total (co- plus cross-polarized) signal components with the far-range telescope. The reasons for this specific design is explained below.

The near-range receiver part was not designed for polarization-sensitive lidar return observations. A 50 mm fiber-wired

telescope collects the total (co- plus cross-polarized) backscatter signals. To realize the dual-FOV polarization lidar technique, we installed another receiver unit (with 50 mm telescope) that permits measurements of the cross-polarized signal at 532 nm at $\mathrm{FOV_{out}} = 2.0$ mrad (top part, red frame, in Fig. 1). The details to the optical elements and the design of this cross-polarized channel is described by Jimenez et al. (2019). Only a polarizer and a collimating lense is used to collect the backscattered cross-polarized laser photons which are then counted by a photomultiplier.

## 3.2 Determination of calibrated depolarization ratios at two FOVs

As mentioned, all Polly instruments measure the cross-polarized and the total (co- plus cross-polarized) signal components. The co-polarized signal component is not recorded. The measurement of the total backscatter cofficient facilitates the determination of the particle backscatter coefficient and, more important, guarantees a direct observation of the extinction-to-backscatter ratio without introducing uncertainties by composing the total backscatter signal from the two (cross and co-polarized) signal

components, measured with different receiver channel efficiencies which need to be measured on a regular basis. To be widely in line with the notation in Engelmann et al. (2016), we switch from indices $\perp$, $\parallel$, and $\perp + \parallel$ (for the total backscatter signal) to c (cross), p (parallel) and t (total), respectively. Because of measuring the cross-polarized and total backscatter signal components, we introduce

$$\delta'(z) = \frac{S_\mathrm{c}(z)}{S_\mathrm{t}(z)} \tag{1}$$

to distinguish this ratio from the volume depolarization ratio as defined by Eq. (9) in part 1. According to Engelmann et al. (2016), the volume depolarization ratio is given by

$$\delta(z) = \frac{1 - \delta'(z)/C}{\delta'(z)F_\mathrm{t}/C - F_\mathrm{c}} \tag{2}$$

with the transmission ratio $F$ and the absolute calibration parameter $C$ (explained below). The transmission ratio $F$ is defined as

$$F_i = \frac{\eta_{i,\mathrm{c}}}{\eta_{i,\mathrm{p}}} \tag{3}$$

with channels $i = \mathrm{c}$ (cross-polarized signal) and $i = \mathrm{t}$ (total backscatter signal). As mentioned, index p indicates here the plane of laser polarization (parallel-polarized signal channel). $F_i$ describes the ratio of transmission $\eta$ for cross-polarized light to





the transmission for co-polarized light for channel $i$. For the Polly system at Punta Arenas (used here), the $F_i$ values were determined from measurements with an artificial light source with a polarizer mounted in front of each channel (Mattis et al., 2009). The values are $F_t = 1.09$ and $F_c = 800$ for the far-range channels (FOV$_{in}$), and $F_t = 1$ and $F_c = 500$ for the near-range channels (FOV$_{out}$).

The absolute calibration parameter $C$ in Eq. (2) is defined as:

$$C = \frac{1 + F_t}{1 + F_c} \sqrt{\delta'(z)_{45°} \, \delta'(z)_{-45°}} \tag{4}$$

and obtained from regular and automated clear-sky measurements. The so-called $\Delta 90°$ calibration method (formerly known as $\pm\Delta 45°$ calibration) (Freudenthaler et al., 2009; Freudenthaler, 2016; Engelmann et al., 2016) is applied to obtain highly accurate depolarization ratio observations for the FOV$_{in}$ channels (far-range receiver). In order to include this method to the

automated measurement procedure of Polly, a remote-controlled rotary mount with a so-called sheet polarizer close to the focal plane of the receiver telescope was added to the system. This sheet polarizer is equipped with an off-center hole to measure without the polarizer into the light path in normal mode by rotating the hole onto the the optical axis. Three times per day, the polarizer is rotated automatically under $-45°$ and $45°$ with respect to the laser polarization plane in the light path for calibration to determine the signal ratios $\delta'(z)_{-45°}$ and $\delta'(z)_{45°}$. The resulting profile of $C$ slightly varies with height because of signal

noise and slightly different conditions during the measurment periods with $-45°$ and $45°$ polarization. Thus, most favorable conditions are cloud-free, clear sky periods for the measurement of $C$. In practice, profile values over several kilometers in the vertical column are averaged to reduce the impact of signal noise on $C$.

    The uncertainties in the measurements and data analysis to obtain the volume depolarization ratio are caused, e.g., by the influence of laser linear polarization purity and uncertainties in the determination of the transmission ratios $F_i$, and the

procedure to obtain the absolute calibration constant $C$. The uncertainties are discussed by Engelmann et al. (2016) and Belegante et al. (2018).

    In our approach of a dual-FOV polarization lidar, we have to distinguish between measurements with FOV$_{in}$ and FOV$_{out}$. Above, we described the retrieval of the volume depolarization ratio for FOV$_{in}$. To indicate this we specify: $C = C_{in}$, $\delta = \delta_{in}$, and $F_i = F_{i,in}$. As in the case of $F_i = F_{i,in}$, we can measure $F_{i,out}$ for FOV$_{out}$ as well.

The calibration constant $C_{out}$ cannot be measured with the rotating polarizer. $C_{out}$ is obtained under the assumption that the volume depolarization ratios for FOV$_{in}$ and FOV$_{out}$ are equal under clear sky conditions (i.e., in the absence of any cloud layer and related multiple scattering effects). It can be shown that for $\delta_{out}(z) = \delta_{in}(z)$

$$C_{out} = \delta'_{out}(z) \left( \frac{1 + F_{t,out} \delta_{in}(z)}{1 + F_{c,out} \delta_{in}(z)} \right). \tag{5}$$

    After careful determination of the $F_i$ and $C$ values for FOV$_{in}$ and FOV$_{out}$ we can now proceed to analyze cloud observations,

as decribed in Sect. 4 in part 1 (Jimenez et al., 2020).

    The cloud-integrated volume depolarization ratio for FOV$_{in}$ signals as defined by Eq. (25) in part 1 is now given by:

$$\bar{\delta}_{in}(z_{bot}, z_{ref}) = \frac{1 - \bar{\delta}'_{in}/C_{in}}{\bar{\delta}'_{in} F_{t,in}/C_{in} - F_{c,in}} \tag{6}$$



with $\overline{\delta}'_{\mathrm{in}}$ calculated from the cross-polarized and the total signal components,

$$\overline{\delta}'_{\mathrm{in}} = \frac{\int_{z_{\mathrm{bot}}}^{z_{\mathrm{ref}}} S_{\mathrm{c,in}}(z)dz}{\int_{z_{\mathrm{bot}}}^{z_{\mathrm{ref}}} S_{\mathrm{t,in}}(z)dz}. \tag{7}$$

For $\mathrm{FOV_{out}}$, we obtain correspondingly

$$\overline{\delta}_{\mathrm{out}}(z_{\mathrm{bot}}, z_{\mathrm{ref}}) = \frac{1 - \overline{\delta}'_{\mathrm{out}}/C_{\mathrm{out}}}{\overline{\delta}'_{\mathrm{out}} F_{\mathrm{t,out}}/C_{\mathrm{out}} - F_{\mathrm{c,out}}} \tag{8}$$

with $\overline{\delta}'_{\mathrm{out}}$ calculated from the cross-polarized and the total signal components,

$$\overline{\delta}'_{\mathrm{out}} = \frac{\int_{z_{\mathrm{bot}}}^{z_{\mathrm{ref}}} S_{\mathrm{c,out}}(z)dz}{\int_{z_{\mathrm{bot}}}^{z_{\mathrm{ref}}} S_{\mathrm{t,out}}(z)dz}. \tag{9}$$

The cloud-integrated depolarization ratios $\overline{\delta}_{\mathrm{in}}(z_{\mathrm{bot}}, z_{\mathrm{ref}})$ and $\overline{\delta}_{\mathrm{out}}(z_{\mathrm{bot}}, z_{\mathrm{ref}})$ and the ratio $\overline{\delta}_{\mathrm{rat}} = \overline{\delta}_{\mathrm{in}}/\overline{\delta}_{\mathrm{out}}$ as defined by Eqs. (25)-(27) in part 1 are the input in the retrieval of cloud microphysical properties as described in Sect. 4 in part 1 and summarized in Table 1.

## 3.3   DACAPO-PESO and LACROS

The lidar observations at Punta Arenas (53.2°S, 70.9°W, 9 m above sea level, a.s.l.), Chile, were conducted in the framework of the DACAPO-PESO campaign from November 2018 to the end of 2020. DACAPO-PESO belongs to a series of long-term ACI-related field studies performed with the mobile LACROS station. Before, we deployed LACROS for the 17-month field campaign CyCARE (Cyprus Clouds, Aerosol and Rain Experiment) at Limassol, Cyprus (October 2016 to March 2018) (Bühl

et al., 2019; Ansmann et al., 2019) in the highly polluted and dusty Eastern Mediterranean. All these campaigns aim at the central question: How do aerosol particles influence the evolution and microphysical properties of liquid-water, mixed-phase and ice clouds and precipitation in different meteorological regimes and at contrasting levels of anthropogenic and natural aerosol concentrations? The novel dual-FOV polarization lidar fills an important gap and covers the ACI research in the case of liquid-water clouds.

We upgraded meanwhile several Polly instruments to dual-FOV polarization lidars. Besides the lidar at Punta Arenas, another upgraded dual-FOV Polly was operated at the North Pole (aboard the German ice breaker Polarstern in the framework of the MOSAiC campaign (September 2019 - September 2020). MOSAiC (Multidisciplinary drifting Observatory for the Study of Artcic Climate, overview and science, https://mosaic-expedition.org) is the largest Artic field campaign ever realized. A new Polly, designed as dual-FOV polarization lidar from the beginning, is now operated at Dushanbe, Tajikistan, in a dusty and

polluted region of Central Asia in the framework of a long-term (unlimited) CADEX follow-up campaign (since June 2019). CADEX (Central Asian Dust Experiment) (Hofer et al., 2017, 2020) was conducted from March 2015 to August 2016. In the framework of the 7-year project EXCELSIOR (*EX*cellence Research *C*enter for *E*arth Survei*L*liance and *S*pace-Based Mon*I*toring *O*f the Envi*R*onment, https://excelsior2020.eu/the-project/), we will deploy a new dual-FOV Polly, integrated in the new EXCELSIOR supersite at Limassol, in the summer of 2020. Finally, the fourth new dual FOV Polly will be setup at

Cabo Verde (in the summer of 2021) in the outflow regime of African dust and biomass burning smoke, as part of ACTRIS.



The mobile Leipzig Cloudnet supersite LACROS (Bühl et al., 2013, 2016; Ansmann et al., 2019) was run continuously at the University of Magallanes (UMAG) at Punta Arenas and covered two summer and winter seasons of aerosol and cloud observations during the DACAPO-PESO campaign. LACROS is equipped with the dual-FOV polarization lidar, a wind Doppler lidar, 35 GHz Doppler cloud radar, ceilometer, disdrometer, and microwave radiometer. In addition, an Aerosol Robotic Network

(AERONET) sun photometer (AERONET, 2020; Holben et al., 1998) was operated.

## 4 Measurements

We discuss two measurement cases of the DACAPO-PESO campaign. The first case study (22 March 2019) deals with the development of an extended altocumulus field in the pristine free troposphere over Punta Arenas, Chile, during the autumn season. The full aerosol and cloud data analysis scheme is applied, the uncertainties in the cloud products obtained with the

dual-FOV lidar are discussed, and the basic results (cloud extinction coefficient, effective radius) are compared with alternative independent retrievals. On 23 February 2019 (case 2), a long-lasting evolution of a stratocumulus deck at the top of the convective summertime boundary-layer was observed. This case is used to illuminate the full potential of a dual-FOV polarization lidar regarding ACI studies in the case of liquid-water clouds.

### 4.1 Case study of 22 March 2019

Figure 2 provides an overview of the cloud conditions over Punta Arenas on 22 March 2019. The narrow FOV signal channels of the Polly lidar are used here. A complex layering of low-level liquid-water clouds, mid-level mixed-phase and upper tropospheric ice clouds was found on this autumn day. In Fig. 2a, optically thin, transparent ice clouds prevailed at heights above about 5 km height, whereas optically thick liquid and mixed-phase clouds, indicated by dark blue columns above the clouds in Fig. 2a dominated at heights below 4 km. The depolarization ratio was high with values of about 0.4 in the ice clouds caused by

strong light depolarization by hexagonal ice crystals. Cloud droplets dominate light-depolarization in the liquid-water clouds at heights below 4 km. The depolarization ratio monotonically increase from values around zero (for ideal spheres) to values around 0.15-0.2 caused by strong multiple scattering by water droplets. It should be mentioned that all POLLY instruments are titled to an off-zenith angle of 5° to avoid a strong impact of specular reflection by falling, horizontally aligned ice crystals which lead to rather low depolarization ratios and, in this way, considerably disturb cloud observations and the separation of

liquid-water, mixed-phase and ice clouds layers.

The results of the Cloudnet classification (Cloudnet, 2020) in Fig. 3a are in good agreement with Fig. 2. The Cloudnet identification and classification method is based on cloud radar, microwave radiometer, and ceilometer observations (Illingworth et al., 2007; Bühl et al., 2016; Baars et al., 2017). According to the Cloudnet classification the clouds below 3.5 km height were mostly liquid-water clouds (blue layers), partly mixed-phase clouds in the height range from 4-6 km, and pure ice clouds

(yellow layers) higher up. The 0°C and −10°C temperature levels were observed at about 2 km and 3.5 km height according to the radiosonde launched at Punta Arenas on 22 March 2019, 12 UTC. Some artifacts are visible. For example, the detection



of ice crystals in the liquid layers at 3-3.5 km around 6 UTC and after 9 UTC is wrong and caused by missing ceilometer observations. The ceilometer laser beam could not penetrate the lower, optically dense cloud layer around 6 UTC.

Figure 3b shows the respective dual-FOV polarization lidar observations. These measurements widely confirm the Cloudnet classification results. The observations are shown in terms of the ratio $\delta_{\mathrm{rat}} = \delta_{\mathrm{in}}/\delta_{\mathrm{out}}$ with $\mathrm{FOV_{in}}= 1$ mrad and $\mathrm{FOV_{out}}=2$ mrad.

It can be seen that the upper layers contained ice crystals which produce a strong, rather narrow, and non-depolarizing forward scattering peak so that both channels ($\mathrm{FOV_{in}}$, $\mathrm{FOV_{out}}$) measure the same backscattering and depolarization features so that $\delta_{\mathrm{rat}}$ was mostly close to 1.0 (reddisch colors). In contrast, $\delta_{\mathrm{rat}}$ was clearly <1.0 in the shallow altocumulus layers between 3 and 3.5 km height (yellow and green color) caused by a larger contribution to the depolarization ratio by droplet multiple scattering in the case of the larger receiver $\mathrm{FOV_{out}}=2$ mrad. Even the presence of drizzle droplets at 1.5 km is detected in Fig. 2b. These

large droplets cause $\delta_{\mathrm{rat}}$ values close to 1 because of a narrow forward scattering peak and similar multiple scattering effects in both FOVs.

In the following, we concentrate on the liquid-water cloud layer from 3-3.5 km height observed over several hours from about 4:30 UTC (1:30 local time) to 11 UTC (8:00 local time). The results of the dual-FOV polarization lidar measurements are shown in Fig. 4. The data analysis procedure was as follows: In the first step, the background and range-corrected total

backscatter signals, available with 30 s temporal resolution, were used to obtain the information about cloud base height $z_{\mathrm{bot}}$. These signals were normalized to the maximum signal value $P_{\mathrm{norm}}(z)$ of total backscatter profile (in the lower part of the cloud layer). A threshold $P_{\mathrm{norm}}(z) > 0.06$ was set to estimate the cloud base height within the signal profile segment from below the cloud base up to about 100 m within the cloud layer. To avoid the influence of signal noise in the cloud base calculation, a smoothing over 5 height bins (37.5 m) was applied to the 30-seconds profiles. This smoothing was only performed for the

determination of $z_{\mathrm{bot}}$. The approach is similar to the method of Donovan et al. (2015) to determine $z_{\mathrm{bot}}$ of liquid-water clouds. The time series of the estimated cloud base height is shown in Fig. 4a.

In the next step, the cross-polarized and total signal components from cloud base to 75 m (10 range or height bins) above cloud base were averaged. A temporal resolution of 2 minutes was selected Fig. 4. The integrated depolarization ratios $\overline{\delta}_{\mathrm{in}}$ and $\overline{\delta}_{\mathrm{out}}$ were calculated by using Eqs. (6)-(9) (in Sect. 3.2). Then, we followed the data analysis strategy as illustrated in Fig. 8

in part 1 and summarized in Table 1. We used $\overline{\delta}_{\mathrm{rat}} = \overline{\delta}_{\mathrm{in}}/\overline{\delta}_{\mathrm{out}}$ for $\Delta z_{\mathrm{ref}}=75$ m to determine $R_{\mathrm{e}}(z_{\mathrm{ref}})$ by means of Eq. (27) (part 1), and afterwards, $\overline{\delta}_{\mathrm{in}}$ and $R_{\mathrm{e}}$ to determine $\alpha(z_{\mathrm{ref}})$ with Eq. (28). By means of the cloud extinction coefficient and the droplet effective radius, we finally obtained the liquid-water content $w_{\mathrm{l}}(z_{\mathrm{ref}})$ with Eq. (4) and the droplet number concentration $N_{\mathrm{d}}(z_{\mathrm{ref}})$ with Eq. (6), again in part 1.

As can be seen in Fig. 4 the shallow altocumulus field which developed in the pristine free troposphere in the Punta Arenas

area showed cloud extinction coefficients around 20 km$^{-1}$ (75 m above clouds base) and droplet effective radii of initially 10 $\mu$m and later on around 7 $\mu$m. The liquid-water content was around 0.1 g m$^{-3}$ and the cloud droplet number concentrations increased from initially of 30 cm$^{-3}$ to 50-80 cm$^{-3}$ later on. The found properties are typical for stratiform cloud layers (stratocumulus, altocumulus) in the marine environment (Miles et al., 2000; Revell et al., 2019).

In Fig. 5, the uncertainties in the cloud retrieval products are shown. The impact of the different error contributions dis-

cussed in Sect. 5 of part 1 are given. The influence of uncertainties in the measured depolarization ratio profiles for the two





FOVs computed by using Eqs. (30) and (34) in part 1 are in general small (on the order of $<5\%$ relative error). The retrieval uncertainties caused by an error in the estimate of the cloud base (CB) height $z_{\mathrm{bot}}$ ($\pm15$ m uncertainty or 10-15% relative error, see Eq. (32) and (36) in part 1) and caused by the theoretical (methodological) aspects (8-15% relative error, see Eqs. (31) and (35) in part 1) dominate the overall uncertainties in the products. For both, the cloud extinction coefficient $\alpha(z_{\mathrm{ref}})$ and droplet

effective radius $R_{\mathrm{e}}(z_{\mathrm{ref}})$ the overall uncertainty is on the order of 15-25%. The uncertainty in the $N_{\mathrm{d}}(z_{\mathrm{ref}})$ value depends on the uncertainties in $\alpha(z_{\mathrm{ref}})$ and $R_{\mathrm{e}}(z_{\mathrm{ref}})$ and is on the order of 50% according to the law of error propagation as indicated by a 50% uncertainty bar in Fig. 4e.

    As discussed by Schmidt et al. (2013, 2014) a bias can be introduced when backscatter signals during periods with varying cloud base height resulting from up and downward motions are averaged. Then, in the lowest part of the cloud, signals from

cloud-free and cloudy air parcels may be averaged which causes this bias. Such an effect cannot be excluded when using the dual-FOV Raman lidar method where signals over 10-30 minutes must be averaged before cloud microphyscial properties can be derived. However, the high temporal resolution now achievable with new dual-FOV polarization technique is of advantage in this respect so that we assume that related uncertainties are small.

    Another uncertainty aspect arises from the fact that we observe the cloud layers with two different field of views (i.e., with

two different eyes) and, thus, monitor two different portions (or cross sections) of the cloud in the horizontal plane. Our method assumes horizontally homogeneous cloud conditions, so that the multiple scattering effect is the only reason for differences in the measurements at the two different FOVs. However, in reality horizontal variations in terms of droplet number concentration, size distribution, and cloud extinction coefficient always occurr, and can, in prinicple, affect the quality of the cloud retrieval products. We checked this potential impact by correlating the separately measured values of depolarization ratios for $\mathrm{FOV}_{\mathrm{in}}$ and

$\mathrm{FOV}_{\mathrm{out}}$. The scatter in the data was very low and did not indicate any significant influence of horizontal cloud inhomogeneities on the ratio $\overline{\delta}_{\mathrm{rat}}$ from which the effective radius of the cloud droplets is retrieved. The necessary signal averaging over 30 s to 120 s may smooth out most of the existing inhomogeneities so that the overall impact is further decreased. A good sign for a negligible impact of horizontal fluctuations in the cloud properties is finally when the time series of the derived values for the different cloud parameters show a coherent behavior.

Figure 6 shows comparisons of our solutions for the cloud extinction coefficient and droplet effective radius with respective results obtained with the single-FOV (SFOV) polarization lidar method (Donovan et al., 2015) and a technique solely based on cloud radar observations of the radar reflectivity factor (Frisch et al., 2002). Here, we used our 35 GHz cloud radar measurements simultaneously conducted at Punta Arenas. The SFOV lidar method is based on cloud simulations with a Monte-Carlo multiple scattering model (Donovan et al., 2010) and, as a result of the lidar simulations, on computed look-up tables of the

cross- and co-polarized signal strengths as a function of cloud microphysical properties. In the case of the Polly instruments, the co-polarized signal is given by the difference of the total minus the cross-polarized signal. The SFOV technique searches for the optimum solution of cloud microphysical properties (cloud extinction coefficient, droplet effective radius) which are consistent with the measured height profiles of the co-polarized and cross-polarized lidar backscatter signals. The products are given as mean values for the lowest 100 m of the liquid-water cloud layer. For the comparison, we used our solutions for 75

m above cloud base and the profile structures shown in Fig. (4) in part 1 to compute the respective mean values of $\alpha$ and $R_{\mathrm{e}}$



for the lowest 100 m within the cloud layers. Note that we started from the SFOV lidar approach to develop the dual-FOV (DFOV) polarization lidar technique. One advantage of the SFOV polarization technique is that it can directly be used by widely distributed polarization lidars (with one FOV). The technique however requires a complicated treatment of the lidar data to perform the retrieval. On the other hand, the dual-FOV polarization technique allows a much more straight forward

retrieval, exploiting the direct relationship between $\bar{\delta}_{\mathrm{rat}}$ and $R_{\mathrm{e}}$.

As can be seen in Fig. 6a, the SFOV polarization lidar slightly underestimates the effective radius of the droplets compared to the other two methods. We used the 1-mrad FOV channel here, as commonly used by wide spread polarization lidars. If we use the 2-mrad FOV channel which is more sensitive to depolarization features caused by multiple scattering in water clouds the agreement with the SFOV approach may improve.

The good agreement of our results with the respective cloud radar solution in Fig. 6b corroborates the quality (accuracy) of our retrieval products. The radar method simply uses the high correlation between radar reflectivity factor and effective radius. In this retrieval procedure, a lognormal droplet size distribution is assumed and the cloud droplet number concentration $N_{\mathrm{d}}$ and the width of the size distribution are needed as input parameters. However, the dependence of the solutions on these input parameters are weak as the solution with different input values indicate. In Fig. 6, we assumed $N_{\mathrm{d}} = 100$ cm$^{-3}$ (as a typical

value for liquid-water clouds) and a logarithmic width of 0.29 as reported for marine stratocumulus over the Southern Ocean (Martin et al., 1994).

The agreement between the SFOV and DFOV solutions is very good in the case of the cloud extinction coefficient. The SFOV polarization lidar technique is obviously robust enough to retrieve the cloud extinction coefficient with good accuracy from the measured cloud depolarization ratio values. This observation is also consistent with Fig. 7 of Donovan et al. (2015).

The retrieved extinction coefficient was found to be not very sensitive to depolarization ratio calibration errors, in contrast to the retrieved values of the effective radius of the droplets.

### 4.1.1 Aerosol and CCN conditions

Fig. 7 shows the aerosol conditions determined from the lidar observations at $\mathrm{FOV}_{\mathrm{in}} = 1$ mrad for this cloud event. We analyzed the altocumulus-free period from 7:45 to 8:45 UTC (see Fig. 2). It was still dark during the time so that we could use the Raman

lidar option to determine the height profiles of the particle extinction coefficient at 355 and 532 nm wavelength (Baars et al., 2016; Hofer et al., 2017).

As can be seen in Fig. 7a, the particle extinction coefficient $\alpha_{\mathrm{par}}$ at 532 nm was in the range of 5-10 Mm$^{-1}$ for the height range from 1.5 to 2.5 km height and even lower at cloud base at 3 km of the cloud layer developing after this cloud-free period. The lidar-derived 532 nm aerosol optical thickness (AOT) was low with values of 0.04 and in consistency with the 500 nm AOT

of 0.03 from the AERONET sun photometer observations on 22 March 2019 after sunrise (13:00-16:00 UTC) (AERONET, 2020).

The different relative humidity profiles in Fig. 7c indicated a comparably low relative humidity (<70%) and thus a low particle water-uptake effect (Skupin et al., 2016; Haarig et al., 2017) in the height range from 1.6–2.2 km marked by dashed lines in Fig. 7. Our extinction-to-$N_{\mathrm{CCN}}$ conversion model, described in Sect. 6 in part 1 (Jimenez et al., 2020) (see also Ta-



ble 1) is applicable for these conditions. By assuming pure marine conditions and sea salt particles as CCN the conversion yields $N_{\mathrm{CCN}}$=40 cm$^{-3}$ for an assumed water supersaturation of 0.2% during droplet formation at cloud base. Such low supersaturation values correspond to the occurrence of weak updrafts with vertical velocities of about 20 cm s$^{-1}$ at cloud base. By applying the continental fine-mode aerosol model, we obtain 160 cm$^{-3}$ assuming urban haze or fire smoke conditions in
the lower free troposphere above Punta Arenas. As shown in Fig. 3e, the cloud droplet number concentration $N_{\mathrm{d}}$ ranged from 50-100 cm$^{-3}$.

HYSPLIT backward trajectories (Hybrid Single Particle Lagrangian Integrated Trajectory Model) (Stein et al., 2015; Rolph et al., 2017; HYSPLIT, 2020), not shown here, indicated westerly winds from the southern Ocean during the last five days before the air mass crossed the lidar station. According to the Ångström exponent in Fig. 7b, describing the wavelength
dependence of the extinction coefficient in the short-wavelength range (355 to 532 nm), traces of continental aerosol (haze or smoke) may have been present in the free troposphere over the lidar site. One option for the occurrence of fine-mode particles could be severe bush fires at the east coast of Australia in February and March of 2019 (Floutsi et al., 2020). The Ångström exponent is defined as $\ln(\alpha_{\mathrm{par}}(\lambda_1)/\alpha_{\mathrm{par}}(\lambda_2))/\ln(\lambda_2/\lambda_1)$ and typically 0.35±0.2 for the 355-532 nm wavelength range, and 0.45±0.2 in the case of the widely used Ångström exponent for the visible-near-IR wavelength spectrum (440-870 nm spectral
range). Such low Ångström exponents clearly below 1 were observed at heights below 1km in Fig. 7b.

However, it is more likely that pure marine conditions prevailed, but that the marine coarse mode particle fraction (large sea salt particles) was widely removed by sedimentation or cloud events in which the large sea salt particles were preferably consumed as CCN and then removed by rainout. As a response of the removal of coarse-mode particles, the Ångström exponent increases. The apparent discrepancy between the low marine $N_{\mathrm{CCN}}$ of about 40 cm$^{-3}$ and the much higher $N_{\mathrm{d}}$ values (80-
100 cm$^{-3}$) is possibly caused by the 0.2% supersaturation assumption in our $N_{\mathrm{CCN}}$ retrieval. The Doppler lidar of LACROS showed the occurrence of gravity wave structures with a pronounced updraft period (45 minutes), from about 8:45 to 9:30 UTC on 22 March 2019 (when a cloud layer formed after the cloud-free period) and vertical winds mostly between 0.5 and 1 m s$^{-1}$, and partly exceeded 1 m s$^{-1}$, so that the water supersaturation was probably clearly higher than 0.5%. At these higher supersaturation conditions the CCN concentration is higher by a factor of about 2 than the value for the 0.2% supersaturation level as
discussed in Mamouri and Ansmann (2016) and recently in Regayre et al. (2019) for Southern Ocean marine CCN conditions. In addition, at such strong vertical winds, even non-sea-salt marine sulfate particles (nss-SO$_4^{2-}$) (Fossum et al., 2020) may have served as CCN. The sulfate-particle-related CCN concentration can be one to two orders of magnitude higher than the sea-salt-particle-related CCN numbers (Fossum et al., 2020). This aerosol species is not considered in the marine conversion model Mamouri and Ansmann (2016) presented in Sect. 6 of part 1.

## 4.2   Case study of 23 February 2019

The second case of the DACAPO-PESO campaign is selected to highlight the significantly improved potential of lidar to contribute to ACI studies with focus on liquid-water clouds. By means of the new dual-FOV polarization lidar technique, cloud and aerosol information can be derived with high temporal resolution which allows us to resolve different phases of cloud



evolution and life cycle and to investigate the impact of individual updrafts on the droplet nucleation rate, droplet growth and corresponding evolution of the effective radius, and the $N_{\mathrm{d}}$–$N_{\mathrm{CCN}}$ relationship in very large detail.

Schmidt et al. (2014, 2015) developed a new strategy to investigate ACI by integrating vertical wind observations with a Doppler lidar. The closest relationship between the number of CCNs below cloud base and the freshly formed cloud droplets

was found during updraft periods. The ACI parameter $E_{\mathrm{ACI},\alpha_{\mathrm{par}}}(N_{\mathrm{d}},\alpha_{\mathrm{par}})$ (see Table 1) and the discussion in Sect. 6 in part 1 was around 0.4 when ignoring the meteorological impact (vertical motion) and about 0.8, when lidar observations exclusively performed during updraft times were considered (Schmidt et al., 2015). However, the data base of Schmidt et al. (2014, 2015) was low (<30 individual cloud cases). When using the dual-FOV Raman lidar technique long signal averaging times are required. More than 200 cloud events were collected, but only in 27 cases the observational constraints were fulfilled (need

for clear skies below the cloud layers over the long signal averaging times of 10-30 min and also a relatively constant cloud base height during these 10–30 minutes). These restrictions were required to avoid biases in the data analysis and thus allow a trustworthy ACI study. All these shortcomings are widely overcome now by using the new dual-FOV polarization lidar technique.

We start with Figure 8a which shows a 6-h cloud measurement at Punta Arenas on 23 February 2019. Liquid-water cloud

parcels permanently formed at the top of the convective summertime planetary boundary layer (PBL) between 15:00 and 21:00 local time (18:00-24:00 UTC). HYSPLIT backward trajectories (not shown) indicated pure marine condition with an airflow from southwest (from the Southern Ocean). GDAS relative humidity (RH) values ranged from 60–65% (at 500m height), 75–85% (at 1000 m height) and were around >95% about 100 m below cloud base during the 18:00–24:00 UTC period (GDAS, 2020). The aerosol particle extinction coefficient $\alpha_{\mathrm{par}}$ decreased with time and indicated a significant reduction

of aerosol particles during the last two hours of the measurement period (22-24 UTC, see Table 2). The particle extinction coefficient is calculated from the measured aerosol backscatter coefficient multiplied by a typical marine extinction-to-backscatter ratio of 25 sr at 532 nm (more details to the determination of backscatter coeffcient profiles below a cloud deck are given in Sect. 6.1 in part 1).

Fig. 8b shows the convective structures of the PBL in terms of the vertical wind component measured with the zenith-

pointing Doppler lidar of LACROS (Cloudnet, 2020). Varying periods with upward (orange) and downward motions (green) were observed. Most updraft velocities were <0.5-0.7 m s$^{-1}$, however some strong updrafts with velocities >1 m s$^{-1}$ occurred as well. The up- and downdrafts modulated cloud formation and cloud base height variations. Water uptake by the aerosol particles at relative humidity >75% (close to cloud base) influenced the strength of the lidar return signals, especially during upwind situations. The dashed lines in Fig. 8a follow the variations of the cloud base height $z_{\mathrm{bot}}$ and indicate 10 further height

levels from 75–750 m below cloud base. These height levels ($z_{\mathrm{aer}}$ in the ACI sketch in Fig. 4 of part 1) are used in the discussion of the results regarding ACI below.

Figure 9 presents the time series of the cloud droplet number concentration $N_{\mathrm{d}}$ for the height $z_{\mathrm{ref}} = z_{\mathrm{bot}} + 75$ m (see Fig. 4 of part 1) obtained from the dual-FOV polarization lidar measurements together with vertical wind indicator (orange for updraft, green for downdraft) and the aerosol proxies $\alpha_{\mathrm{par}}$ and $N_{\mathrm{CCN}}$. The aerosol proxies are mean values for the height range from

375-600 m below cloud base at $z_{\mathrm{bot}}$, thus for the height range from $z_{\mathrm{aer}} = z_{\mathrm{bot}} - 375$ m to $z_{\mathrm{aer}} = z_{\mathrm{bot}} - 600$ m according to



Fig. 4 in part 1. The temporal resolution in Fig. 9 is one minute. The marine aerosol conversion parameterization (Mamouri and Ansmann, 2016) is applied to obtain $N_{\mathrm{CCN}}$ (see Table 1, more explanations are given in Sect. 6 in part 1). This conversion corrects for aerosol water uptake effects for a typical marine RH of 80% and holds for the water supersaturation level of 0.2%. Error bars indicate the retrieval uncertainty of 20% (marine particle extinction coefficient $\alpha_{\mathrm{par}}$) and 50% ($N_{\mathrm{d}}$, $N_{\mathrm{CCN}}$).

Figure 9 shows that the cloud droplet number concentration $N_{\mathrm{d}}$ varied strongly and was clearly correlated with updraft occurrence during the 19:30-24:00 UTC time period whereas the aerosol proxies $N_{\mathrm{CCN}}$ and $\alpha_{\mathrm{par}}$ were likewise smooth functions of time. However, the true or actual $N_{\mathrm{CCN}}$ values probably showed large variations because the CCN level (actually occurring) depends on actual updraft speed and related actually occurring supersaturation level. For updraft velocities of 1 m s$^{-1}$ and corresponding supersaturation exceeding 0.5% $N_{\mathrm{CCN}}$ is approximately a factor of 2 higher than the shown ones for the fixed
0.2% water supersaturation as outlined in Mamouri and Ansmann (2016) and already pointed out in Sect. 4.1.

     On average, $N_{\mathrm{d}}$ ranged from 20-100 cm$^{-3}$ before 21:30 UTC and 10-30 cm$^{-3}$ later on. Peak values exceeded 200 cm$^{-3}$. The aerosol parameters $N_{\mathrm{CCN}}$ and $\alpha_{\mathrm{par}}$, indicating clean conditions (with horizontal visibility >50 km), were mostly in the range of from 30-60 Mm$^{-1}$ before 21:15 UTC and 20-35 Mm$^{-1}$ later on in the case of $\alpha_{\mathrm{par}}$ and >100 cm$^{-3}$ before 21:15 UTC and clearly <100 cm$^{-3}$ later on in the case of $N_{\mathrm{CCN}}$. The decrease of $N_{\mathrm{d}}$ with time is in line with the decrease of $N_{\mathrm{CCN}}$ and $\alpha_{\mathrm{par}}$.
Table 2 summarizes the aerosol and cloud observations and contains mean values of all derived aerosol and cloud properties for two time periods characterized by different aerosol conditions.

     In Fig. 10, the impact of upward and downward motion on the measured cloud properties $N_{\mathrm{d}}$, $R_{\mathrm{e}}$, and $\alpha_{\mathrm{par}}$ are shown. For comparison, we also included the respective variations of the aerosol proxy $N_{\mathrm{CCN}}$. Such correlations of $N_{\mathrm{d}}$ and $R_{\mathrm{e}}$ with vertical velocity at cloud base are new options of combined dual-FOV and Doppler lidar profiling. As can be seen, very clear
correlations were not found. A pronounced influence of updraft speed on $N_{\mathrm{d}}$ and $R_{\mathrm{e}}$ cannot be expected in this case of a long-lived, well-developed stratocumulus cloud deck. At such pre-existing cloud conditions there is a competition between droplet nucleation and water uptake by the existing droplets in case of a given supersaturation. However, the following tendencies are visible in the four correlation plots of Fig. 10. During downdraft periods, $N_{\mathrm{d}}$ decreases, probably caused by droplet evaporation. This removal of small droplets is correlated with a comparable large effective radius $R_{\mathrm{e}}$ for the remaining, surviving droplets.
A weak decrease of the cloud extinction coefficient is visible when going from updraft to downdraft conditions in line with the dissolution of droplets. If the vertical wind vector turns from downdraft to updraft conditions, we see an accumulation of values for the parameters in the vertical velocity range from 0-70 cm s$^{-1}$. The updraft speeds occurred most often. For these weak to moderate updraft velocities $N_{\mathrm{d}}$ and $N_{\mathrm{CCN}}$ were then found in the range from 15-100 cm$^{-3}$ and 75-200 cm$^{-3}$, respectively, and the corresponding activation ratio $N_{\mathrm{d}}/N_{\mathrm{CCN}}$ was mostly between 0.25 and 0.5. New droplet formation and growth of existing
droplets by water uptake led to a slight increase of the cloud extinction coefficient in many cases. A weak reduction of the mean effective radius during upward motions may indicate new droplet nucleation in the presence of existing droplets.

     The found activation ratios are in reasonable agreement with literature values. Revell et al. (2019) reported activation ratios accumulating around 0.5-0.6 for pure marine conditions of the Southern Ocean, The clouds formed at 800 m above the surface. In this model-based study, simulated $N_{\mathrm{CCN}}$ values (for a supersaturation of 0.2%) were in the range from 50-80 cm$^{-3}$ during
the late summer season (February and March) and $N_{\mathrm{d}}$ showed values from 30-50 cm$^{-3}$. According to the recent publication





of Regayre et al. (2019), $N_{\mathrm{CCN}}$ is usually underestimated by a factor of 2 in models focusing on aerosols and clouds in the Southern Ocean. Our lidar-derived $N_{\mathrm{CCN}}$ values are in very good agreement with the CCN numbers presented by (Regayre et al., 2019) of usually 100-200 cm$^{-3}$. These authors constrained their simulations to recent CCN observations aboard a Russian research vessel traveling around entire Antarctica (Schmale et al., 2019). Our findings regarding the activation ratio are also

in reasonable agreement with airborne in situ observations of $N_{\mathrm{d}}$ and $N_{\mathrm{CCN}}$ over the Southeast Pacific stratocumulus cloud regime west of Northern Chile (Zheng et al., 2011; Painemal and Zuidema, 2013). Their activation ratios accumulated from 0.5-0.7 at pristine marine, slightly polluted environmental conditions.

Some caution has to be exercised in the interpretation of the results in Fig. 10 because of the uncertainties in the retrieval products discussed above (case study 1) and because of the assumptions made in the development of the dual-FOV polarization

lidar technique. We assume subadiabatic conditions and corresponding profile structures for the different cloud parameters as shown in Fig. 4 of part 1 for the lowermost 75 m of the cloud layer. We also assume a gamma size distribution to describe the droplet size spectrum. These assumptions may no longer hold for an aged, pre-existing cloud layer (especially not during downdraft periods) in which droplet collision and coalescence processes, entrainment and droplet evaporation takes places. However, the gamma size distribution and subadiabtic cloud conditions were introduced to develop our dual FOV lidar method

with focus on the most interesting scenarios (updraft periods). The new method is primarily based on the strong relationship between the measured ratio $\overline{\delta}_{\mathrm{rat}} = \overline{\delta}_{\mathrm{in}}/\overline{\delta}_{\mathrm{out}}$ and the droplet effective radius $R_{\mathrm{e}}$ and the clear relationship between the depolarization ratio $\overline{\delta}_{\mathrm{in}}$ (for FOV$_{\mathrm{in}}$) and the cloud extinction coefficient $\alpha$ for a given $R_{\mathrm{e}}$ value, known from the first part of the retrieval procedure.

In the next step, we computed the ACI parameter $E_{\mathrm{ACI},\alpha_{\mathrm{par}}}(N_{\mathrm{d}}, \alpha_{\mathrm{par}})$ (see Table 1 and Sect. 6 in part 1 for more expla-

nations). In Fig. 11, the correlation between the derived $N_{\mathrm{d}}$ and measured $\alpha_{\mathrm{par}}$ values (in Fig. 9 and 10) are considered, separately for updraft and downdraft periods. We use the particle extinction coefficient $\alpha_{\mathrm{par}}$ (and not $N_{\mathrm{CCN}}$) in the correlation because this quantity is directly obtained from the lidar observations with a low uncertainty of 20%. Disregarding the aerosol proxy used, we notice a large scatter in the correlated data. This is typical for aerosol and cloud parameters determined in well-developed, pre-existing liquid-water cloud layers (McComiskey et al., 2009). As mentioned above, the large scatter is

caused by the strong variability of $N_{\mathrm{d}}$ (as a function of the varying vertical wind conditions) compared to the low variability in the particle extinction coefficient which is not a function of vertical wind velocity. To obtain $E_{\mathrm{ACI},\alpha_{\mathrm{par}}}(N_{\mathrm{d}}, \alpha_{\mathrm{par}})$ a linear regression analysis is applied to the $\log(N_{\mathrm{d}})$-$\log(\alpha_{\mathrm{p}})$ data field. $E_{\mathrm{ACI},\alpha_{\mathrm{par}}}(N_{\mathrm{d}}, \alpha_{\mathrm{par}})$ is equal to the slope of the regression line. As expected, the aerosol impact on $N_{\mathrm{d}}$ is stronger for upward motions.

As a final task, we applied such correlation studies and regression analysis as presented in Fig. 11 to the full sets of $N_{\mathrm{d}}$, $\alpha_{\mathrm{par}}$

and $N_{\mathrm{CCN}}$ data. We performed regressions analyses with different sets of aerosol proxis for different height levels $z_{\mathrm{aer}}$ below cloud base height $z_{\mathrm{bot}}$ (as illustrated in Fig. 4 in part 1 and indicated by the 10 dashed lines in Fig. 8a) to investigate to what extent water uptake corrupts the ACI study. The result is shown in Fig. 12.

The respective ACI efficiency values $E_{\mathrm{ACI}}$ are assigned to the heights of the aerosol layers (with respect to cloud base) of which the aerosol proxies were considered in the $E_{\mathrm{ACI}}$ computations. This way of presenting the ACI efficiency values allows

us to check the impact of water uptake by the marine particles when the relative humidity steadily increases and reaches 100%





at cloud base. As can be seen, the ACI efficiency $E_{\mathrm{ACI}}$ for well-defined updraft conditions decreases from values close to 1 ( the optimum value for the expected strong impact of marine particles on the droplet number concentration) at heights around 400 m below cloud base to values around 0.5 very close to cloud base. Obviously, water uptake leads to a broadening of the range of observable extinction coefficients. For dry particles, the extinction coefficients varies over a more narrow range so

that the relative increase of the directly measured $\log(\alpha_{\mathrm{par}})$ is proportional to the relative increase in $\log N_{\mathrm{d}}$. This no longer the case when all particles grow by water uptake. Then the increase of $\log(\alpha_{\mathrm{par}})$ is linked to a much lower relative increase of the droplet number concentration (lower by almost a factor of 2). For downdraft periods the decrease of $E_{\mathrm{ACI}}$ with water uptake effects is less clear and pronounced as can be seen in Fig. 11 because of the generally not well-defined link between droplet nucleation and available CCN.

It is interesting to note at the end that Shinozuka et al. (2015) found that the maximum value of $E_{\mathrm{ACI},\alpha_{\mathrm{par}}}(N_{\mathrm{d}},\alpha_{\mathrm{par}})$ can only be about 0.8-0.85, i.e., when the aerosol particle extinction coefficient $\alpha_{\mathrm{p}}$ is used as aerosol proxy. $E_{\mathrm{ACI}}$=0.9-1.0 is only possible when $N_{\mathrm{CCN}}$ is considered as shown in Fig. 12. The reason for this is that $N_{\mathrm{CCN}}$ is proportional to $\alpha_{\mathrm{par}}^{0.85}$ and not to $\alpha_{\mathrm{par}}$ in Eq. (38) in part 1.

## 5 Summary, conclusions, and outlook

In a companion article (Jimenez et al., 2020), we presented a new polarization-lidar-based approach to derive microphysical properties in the lower part of pure liquid-water clouds. Extended simulations were performed regarding the relationship between cloud microphyscial and light-extinction properties and the cloud depolarization ratio measured with lidar at two different FOVs. These simulations served as the basis for the development of the new dual-FOV polarization lidar method. The effective radius of the cloud droplets and the cloud light-extinction coefficient in the lowest 50-100 m of the cloud layer can be

derived with a relative error of 20-25%. From the quantities, the cloud droplet number concentration can be computed with an error of the order of 50%.

In part 2, the new lidar technique was combined with the aerosol polarization lidar method which enables the retrieval of CCN concentrations below cloud base and with Doppler lidar observations of the vertical wind component and thus of updraft and downdraft occurrence at cloud base. We implemented the novel dual-FOV polarization lidar technique into a multiwavelength

polarization Raman lidar (Polly), which is now involved in the long-term DACAPO-PESO field campaign in Punta Arenas, southern Chile, at the southern most tip of South America.

Two case studies were presented. Case 1 was used to discuss the basic and principle features of the new cloud retrieval technique. This case study included an uncertainty discussion and comparisons with alternative approaches to derive cloud microphysical properties such as the single-FOV polarization lidar technique (Donovan et al., 2015) and a cloud-radar based

approach (Frisch et al., 2002). Good agreement was found.

Case 2 highlighted the new and extended potential of lidar to contribute to detailed ACI studies in the case of liquid-water clouds. Profiling of aerosol-relevant aerosol parameters close to cloud base, cloud microphysical properties just above cloud base, and of vertical wind with one-minute resolution was possible and enabled a detailed updraft and downdraft-resolved ACI





study. For typical updraft conditions with vertical velocities <50-70 cm s$^{-1}$ we found $N_\mathrm{d}$ and $N_\mathrm{CCN}$ values (for 0.2% water supersaturation) ranging from 15-100 cm$^{-3}$ and 75-200 cm$^{-3}$, respectively, and corresponding activation ratios $N_\mathrm{d}/N_\mathrm{CCN}$ mostly from 0.25-0.5 in the well-developed, pre-existing stratocumulus deck at the top of the pristine marine boundary layer over Punta Arenas. ACI studies were performed separately for updraft and downdraft conditions with particle extinction coeffi-

cient $\alpha_\mathrm{par}$ as well as with $N_\mathrm{CCN}$ as aerosol proxy. High ACI values of 0.8-1.0 were found. The impact of aerosol water-uptake on the ACI studies was illuminated with the result that the highest ACI values were obtained by considering the aerosol proxies $\alpha_\mathrm{par}$ or $N_\mathrm{CCN}$ measured at heights about 500 m below cloud base (and thus for dry aerosol conditions) in the ACI computations.

As an outlook, we will extend our ACI studies by means of the dual-FOV lidar method. We equipped three further Polly

instruments with the dual-FOV polarization lidar technique. These lidars are or were operated at the North Pole (at 85-90°N) onboard the German ice breaker Polarstern from September 2019 to September 2020, at Dushanbe, Tajikistan, at polluted and dusty conditions in Central Asia since June 2019, and at Limassol, Cyprus, in the polluted and dusty Eastern Mediterranean since summer 2020. A fourth dual-FOV Polly lidar will start long-term monitoring at Mindelo, Cabo Verde, in the outflow regime of pollution and dust from western and central Africa in 2021. A mobile dual-FOV Polly will be moved to New

Zealand for further ACI studies in the Southern Ocean in 2021. All these field activities will be used to characterize ACI in the case of liquid-water clouds at very different aerosol and meteorological conditions.

The integration of the dual-FOV-polarization lidar technique into the LACROS infrastructure can be regarded as a next systematic step to improve the capability of state-of-the-art ground-based remote sensing towards an overall monitoring of aerosol-cloud interaction in liquid-water clouds as presented here and mixed-phase clouds and cirrus layers as presented re-

cently by Bühl et al. (2019) and Ansmann et al. (2019).

*Data availability.* Polly lidar observations (level 0 data, measured signals) are in the PollyNET data base (PollyNET, 2020). LACROS observations (level 0 data) are stored in the Cloudnet data base of LACROS (Cloudnet, 2020). All the analysis products are available at TROPOS upon request (info@tropos.de). Backward trajectories analysis has been supported by air mass transport computation with the NOAA (National Oceanic and Atmospheric Administration) HYSPLIT (HYbrid Single-Particle Lagrangian Integrated Trajectory) model

(HYSPLIT, 2020). AERONET photometer observations of Punta Arenas are in the AERONET data base (AERONET, 2020).

*Author contributions.* CJ and AA prepared the manuscript. CJ developed the new method and analyzed all field campaign observations. CF, RE, and RW upgraded the Polly instrument. CJ, RE, PS, MR, ZY, BB took care of the DACAPO PESO campaign and the field campaign instrumentation. ZY determined the lidar calibration constants (for the aerosol retrieval below cloud base) for the first year of the DACAPO-PESO campaign. DD, AM, PS, JS, and UW supported the discussion and interpretation of the observations.

*Competing interests.* The authors declare that they have no conflict of interest.



*Special issue statement.* This article is part of the special issue "EARLINET aerosol profiling: contributions to atmospheric and climate research".

*Acknowledgements.* The authors wish to thank TROPOS and UMAG for their logistic and infrastructural support during the preparation
5  phase and during the DACAPO-PESO campaign at Punta Arenas. We are grateful to the technicians of the mechanical workshop of TROPOS for the always prompt and careful assistance when upgrading the Polly. We thank AERONET for their continuous efforts in providing high-quality measurements and products. Aerosol sources apportionment analysis has been supported by air mass transport computation with the rather convenient NOAA (National Oceanic and Atmospheric Administration) HYSPLIT (HYbrid Single-Particle Lagrangian Integrated Trajectory) model.

10  *Financial support.* This research was partially funded by the pro-gram DAAD/Becas Chile, grant no. 57144001. This activity is supported by the ACTRIS Research Infrastructure (EU H2020-R&I), grant agreement no. 654109.



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



**Table 1.** Overview of the cloud and aerosol retrieval procedure (step-by-step data analysis). The retrieval procedure starts with the determination of the cloud base height $z_{bot}$. The cloud depolarization ratios $\overline{\delta}_{in}$ and $\overline{\delta}_{out}$ and the ratio $\overline{\delta}_{rat} = \overline{\delta}_{in}/\overline{\delta}_{out}$, integrated over the height range from cloud base at $z_{top}$ to the cloud retrieval or reference height $z_{ref}$, are calculated from the height profiles of measured volume linear depolariaztion ratios (see Sect. 3.2). The cloud products $R_e$, $\alpha$, $w_l$, and $N_d$ are given for the reference height $z_{ref}$, 75 m above cloud base height $z_{bot}$. The computation of the aerosol-cloud-interaction (ACI) efficiencies $E_{ACI}$ is based on $N_d$, particle extinction coefficient $\alpha_{par}$, and CCN concentration $N_{CCN}$ at $z_{aer}$, usually several 100 m below cloud base. The aerosol proxies are determined from aerosol measurements with the same dual-FOV lidar.

| Parameter | Symbol | Eq. (see part 1) | Uncertainty |
|---|---|---|---|
| Cloud base height | $z_{bot}$ | | 0.1-1% |
| Cloud depolarization ratios | $\overline{\delta}_{in}(z_{bot}, z_{ref})$ | Eq. (25) | 5% |
| | $\overline{\delta}_{out}(z_{bot}, z_{ref})$ | Eq. (26) | 5% |
| | $\overline{\delta}_{rat}(z_{bot}, z_{ref})$ | Eq. (27) | 10-15% |
| Droplet effective radius | $R_e(z_{ref})$ | Eq. (28) | 15% |
| Cloud extinction coefficient | $\alpha(z_{ref})$ | Eq. (29) | 15-20% |
| Liquid water content | $w_l(z_{ref})$ | Eq. (4) | 25% |
| Cloud droplet number concentration | $N_d(z_{ref})$ | Eq. (6) | 25-75% |
| Aerosol depolarization ratio | $\delta_{par}(z)$ | | 5-10% |
| Aerosol extinction coefficient | $\alpha_{par}(z_{aer})$ | | 20% |
| Cloud condensation nucleus concentration | $N_{CCN}(z_{aer})$ | Eqs. (38) – (40) | 30-100% |
| Aerosol-cloud-interaction efficiency | $E_{ACI,\alpha_{par}}(N_d, \alpha_{par})$ | Eq. (41) | |
| Aerosol-cloud-interaction efficiency | $E_{ACI,N_{CCN}}(N_d, N_{CCN})$ | Eq. (42) | |

**Table 2.** Mean values and standard deviations of cloud properties observed in the liquid-water cloud layer at 3 km height on 23 February 2019. The aerosol properties for the height range from 375-600 m below cloud base are given in addition. During the selected averaging periods (before and after 22:00 UTC), very different cloud and aerosol properties were found.

| | 18:00-22:00 UTC | 22:00-24:00 UTC |
|---|---|---|
| $N_d$, **cm**$^{-3}$ | $42.4 \pm 46.3$ | $21.4 \pm 15.4$ |
| $R_e$, **µm** | $8.5 \pm 2.7$ | $10.4 \pm 2.7$ |
| $\alpha$, **km**$^{-1}$ | $10.0 \pm 2.3$ | $9.3 \pm 1.8$ |
| $w_l$, **gm**$^{-3}$ | $0.054 \pm 0.015$ | $0.063 \pm 0.02$ |
| $N_{CCN}$, **cm**$^{-3}$ | $110 \pm 21$ | $72 \pm 14$ |
| $\alpha_{par}$, **km**$^{-1}$ | $0.025 \pm 0.006$ | $0.015 \pm 0.003$ |

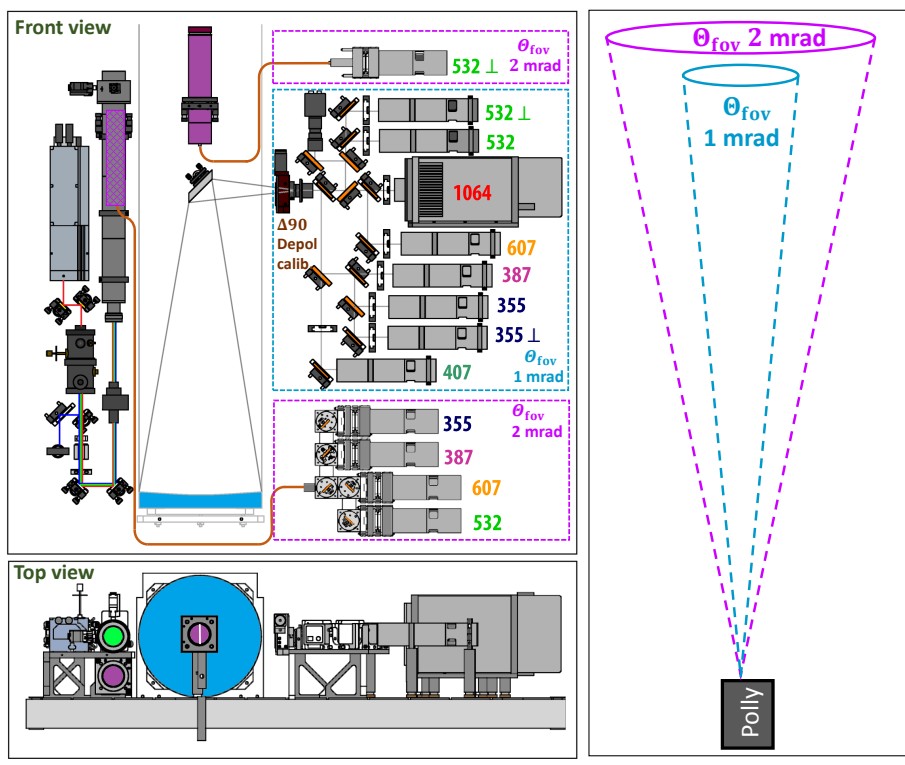

**Figure 1.** Optical setup of the Polly lidar (left side, same as in Fig. 3 of Engelmann et al. (2016)). The upper left part displays the front view of the system, the lower left part a top view. The transmitter unit is mounted to the left of the main lidar telescope (in blue). Laser light transmission is indicated by a green circle in the top view sketch. Backscattered light is collected with a Newtonian telescope (blue area in the top view sketch, FOV=1 mrad, far-range telescope) and then passed towards the far-range receiver unit to the right. All optical elements and detector channels belonging to the 1 mrad FOV receiver block are given in a blue frame. The numbers indicate the wavelength in nanometers of the detection channels and ⊥ denotes the cross-polarized channels. A polarizer is mounted in front of the pinhole (entrance of the far-range receiver unit) and used for the absolute calibration of the depolarization measurements at 1 mrad FOV (for details see Engelmann et al. (2016)). The violet parts and the violet frame (top view, front view sketch) belong to the 2 mrad FOV receiver unit (near-range receiver unit). An additional 5 cm receiver telescope (violet, for details see Jimenez et al. (2019) is mounted above the secondary mirror of the far-range telescope and collects the cross-polarized signal component at FOV=2 mrad. Right to the optical setup, the overall dual-FOV polarization lidar configuration is illustrated highlighting the different FOVs.





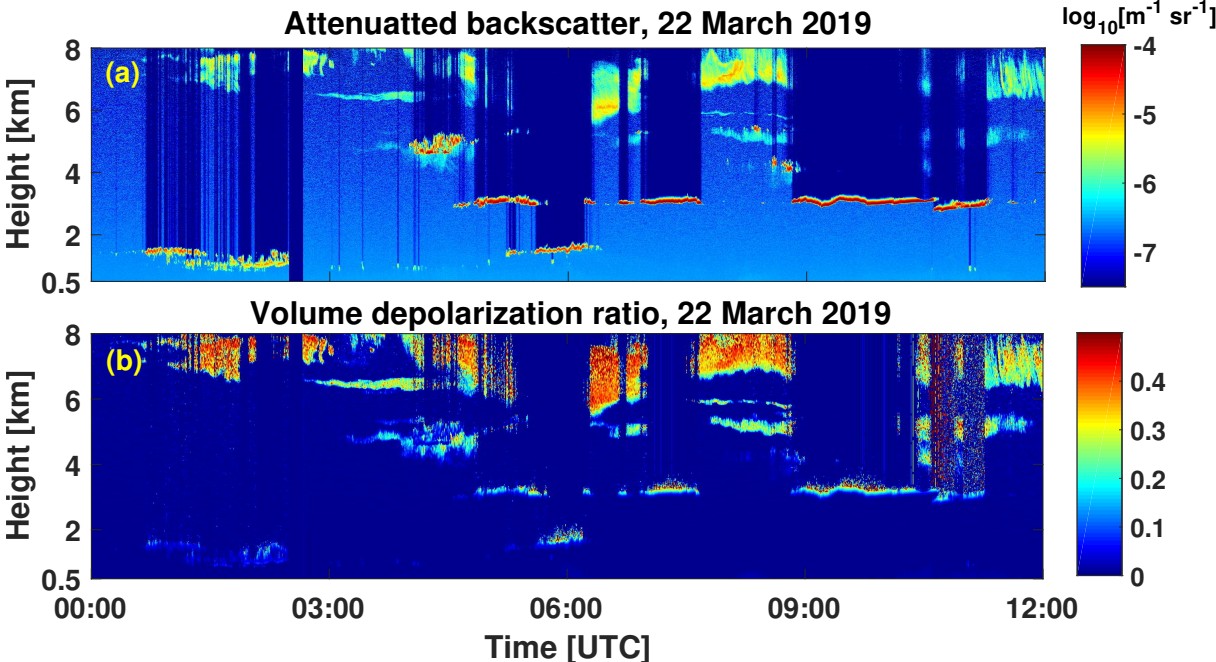

**Figure 2.** Liquid-water cloud layers at heights <3.5 km and ice-containing clouds between 4 and 8 km height in the pristine marine atmosphere over Punta Arenas observed with polarization lidar on 22 March 2019. Height-time display of (a) attenuated backscatter at 1064 nm and (b) volume linear depolarization ratio at 532 nm measured with 30 s temporal and 7.5 m verical resolution are presented.

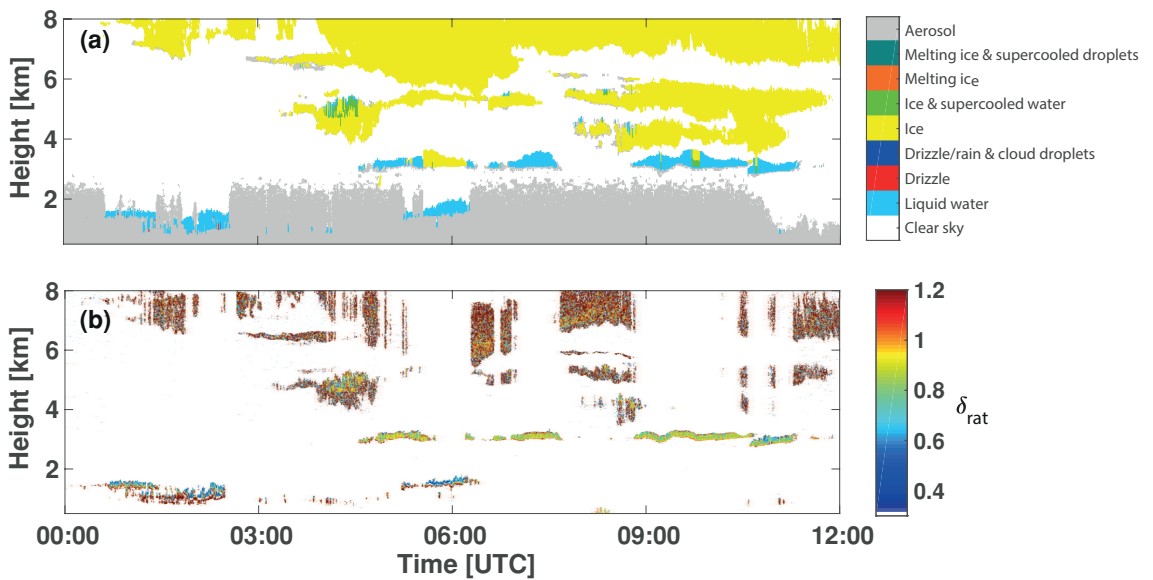

**Figure 3.** (a) Cloudnet target classification of the cloud layers shown in Fig. 2 and (b) ratio $\delta_{\mathrm{rat}} = \delta_{in}/\delta_{out}$ measured with the dual-FOV polarization lidar Polly (FOV$_{\mathrm{in}}$= 1 mrad, FOV$_{\mathrm{out}}$=2 mrad).



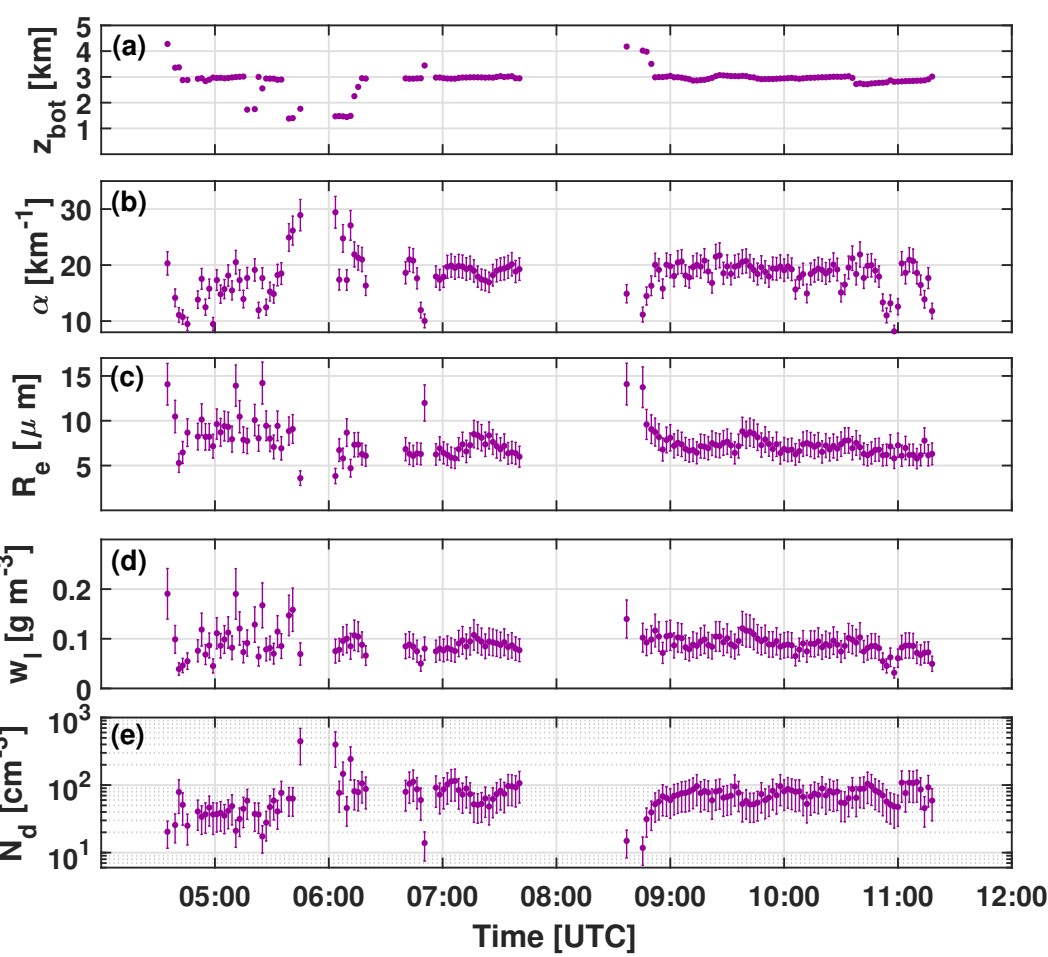

**Figure 4.** Dual-FOV polarization lidar observation of (a) cloud base height $z_{\mathrm{bot}}$ of detected liquid cloud layers, (b) cloud extinction coefficient $\alpha(z_{\mathrm{ref}})$, (c) droplet effective radius $R_{\mathrm{e}}(z_{\mathrm{ref}})$, (d) liquid-water concentration $w_{\mathrm{l}}(z_{\mathrm{ref}})$, and (e) droplet number concentration $N_{\mathrm{d}}(z_{\mathrm{ref}})$ for the liquid-water clouds mostly located between 3.0 and 3.5 km height shown in Fig. 2. $z_{\mathrm{ref}}$ is 75 m above cloud base. Time resolution is 120 s. Error bars indicate the estimated overall uncertainty in the retrieved values.



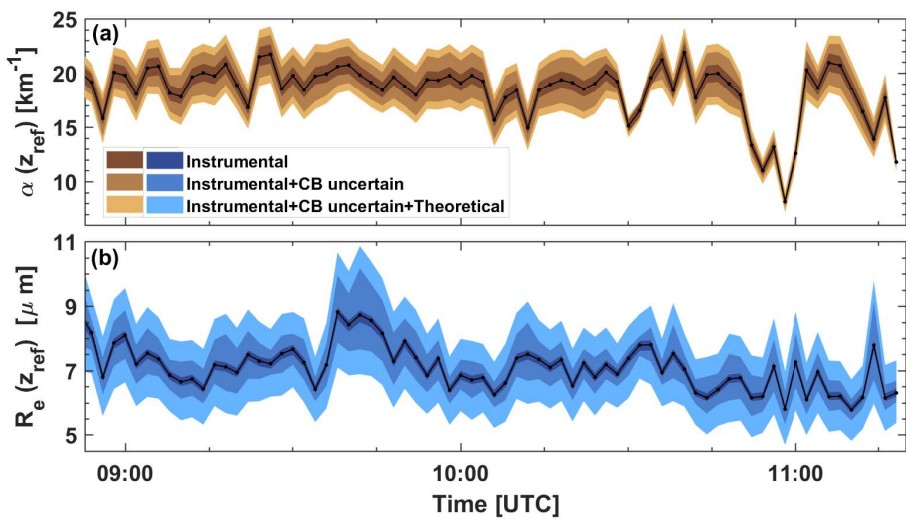

**Figure 5.** Contribution of the different error sources in the retrieval of the cloud extinction coefficient (a) and droplet effective radius (b) shown in Fig. 4. In (a), the impact of instrumental uncertainties is calculated with Eq. (34) in part 1 (Jimenez et al., 2020), the uncertainty in the cloud base (CB) determination with Eq. (36), and of theoretical uncertainties with Eq. (35) in part 1. In (b), the impact of instrumental uncertainty is calculated with Eq. (30), the influence of the CB uncertainty with Eq. (32), and of theoretical uncertainties with Eq. (31). The uncertainty in the CB determination and the methodological (theoretical) uncertainties dominate the overall retrieval uncertainties.



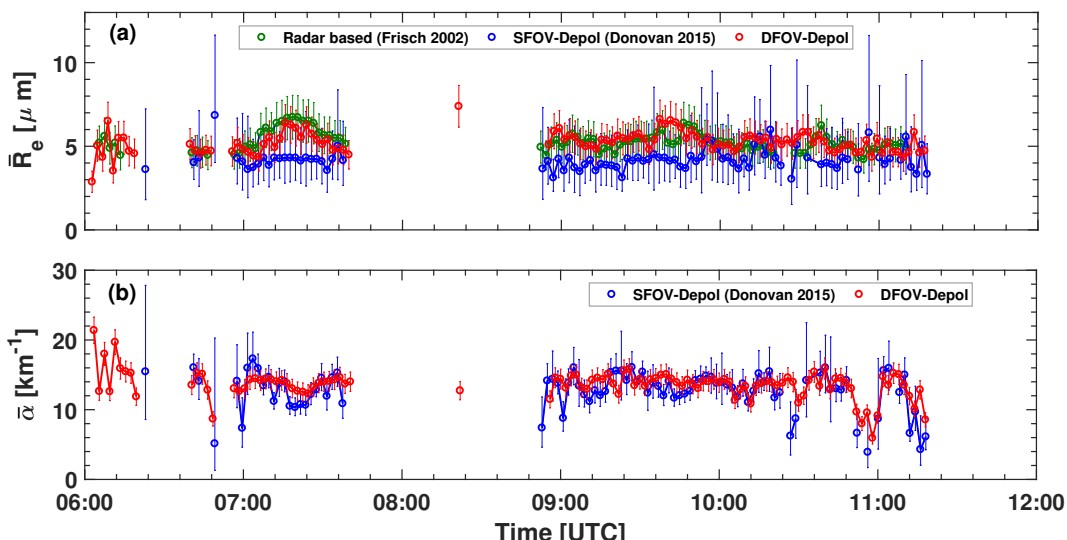

**Figure 6.** Comparison of (a) droplet effective radius values $\overline{R}_e$ (mean values for the lowest 100 m in the liquid-water cloud layer) and (b) respective mean extinction coefficient values $\overline{\alpha}$ obtained with the single-FOV polarization lidar method (SFOV-Depol) (Donovan et al., 2015) and the dual-FOV polarization lidar technique (DFOV-Depol) for the case shown in Fig. 4. In addition, the results (in green) obtained with a cloud-radar approach (Frisch et al., 2002) are shown in (a). Observations of the radar reflectivity factor performed with the LACROS 35 GHz cloud radar at Punta Arenas are used here. Error bars indicate the uncertainty range.



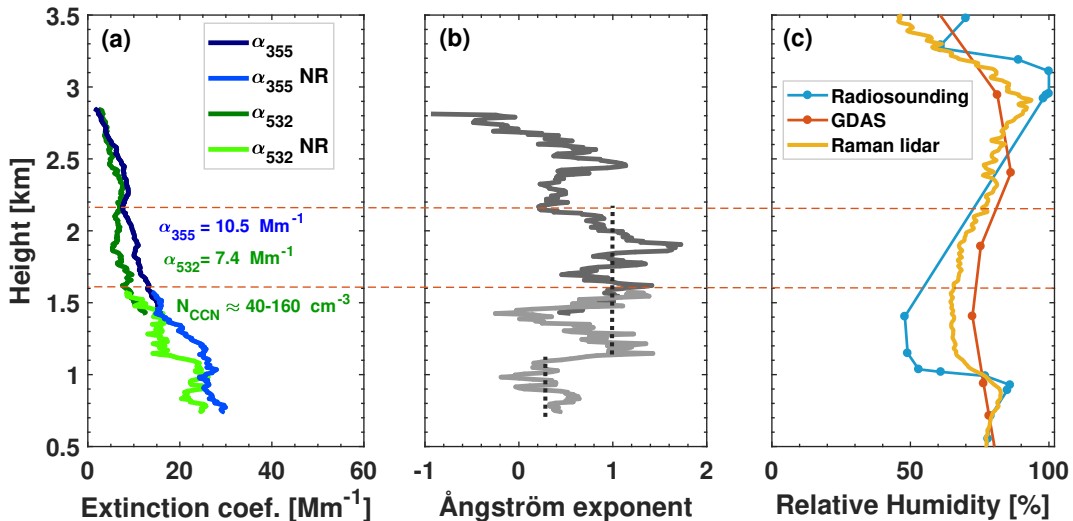

**Figure 7.** Aerosol observation with the dual-FOV polarization Raman lidar on 22 March 2019 during the altocumulus-free period from 7:45-8:45 UTC (see Figs. 2 and 4). (a) Particle extinction coefficient at 355 and 532 nm (NR indicates the determination from Raman signal profiles measured with the near-range of $\text{FOV}_{\text{out}}$ telescope), (b) Ångström exponent (355-532 nm spectral range) computed from the extinction profiles in (a), and (c) relative humidity profiles calculated from the Raman lidar observation of the water vapor mixing ratio (for the time period from 7:45-8:45 UTC) and by using the respective GDAS temperature profile (orange), relative humidity taken from the GDAS data set (red), and as measured with Punta Arenas radiosonde (blue, launched at 12 UTC). The mean values of the particle extinction coefficients $\alpha_{355}$ at 355 nm and $\alpha_{532}$ at 532 nm for the height range from 1.6-2.15 km (driest region, indicated by dashed horizontal lines) are given as numbers in (a). The layer mean 532 nm extinction coefficient is used to derive the $N_{\text{CCN}}$ range by assuming pure marine conditions (minimum value) and pure urban haze conditions (maximum value) and a supersaturation of 0.2% during droplet nucleation events. The dashed vertical lines in (b) indicate different Ångström values for the boundary layer and for the relatively dry part of the free troposphere below the cloud deck.



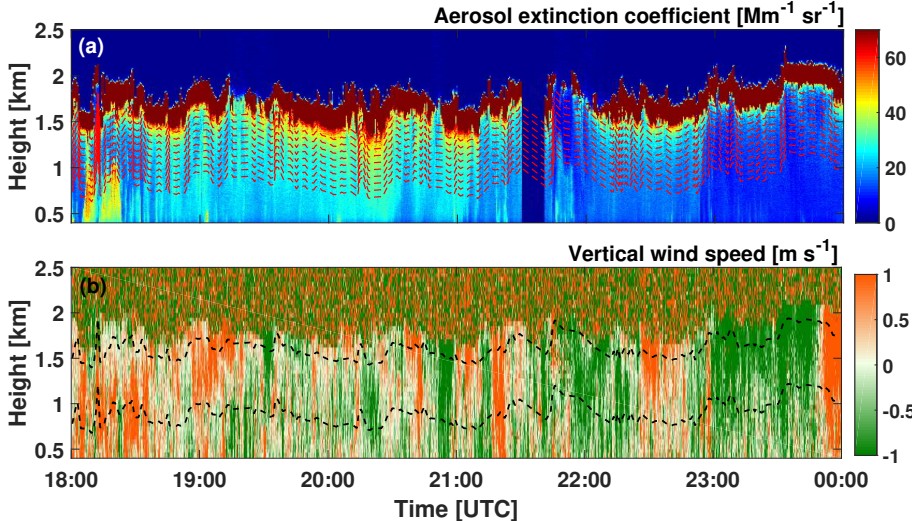

**Figure 8.** Lidar observation of the convective cloud-topped planetary boundary layer in the afternoon and evening of the summer day of 23 Febrruray 2019. (a) Aerosol extinction coefficient (blue to yellow colors) up to the base of the stratocumulus layer (dark red) at around 1500 m height, and (b) vertical wind component (orange: upward motion, green: downward motion) measured with the zenith-pointing Doppler lidar of the LACROS facility. The cloud base strongly varies with the permanently changing updraft and downdraft conditions. Red dashed curves in (a) show height levels of constant distance of 75 to 750 m from cloud base. the black dashed lines in (b) show the cloud base height $z_{\mathrm{bot}}$ and the height level 750 m below cloud base. For these height levels, aerosol proxies for the ACI studies are computed as discussed below.

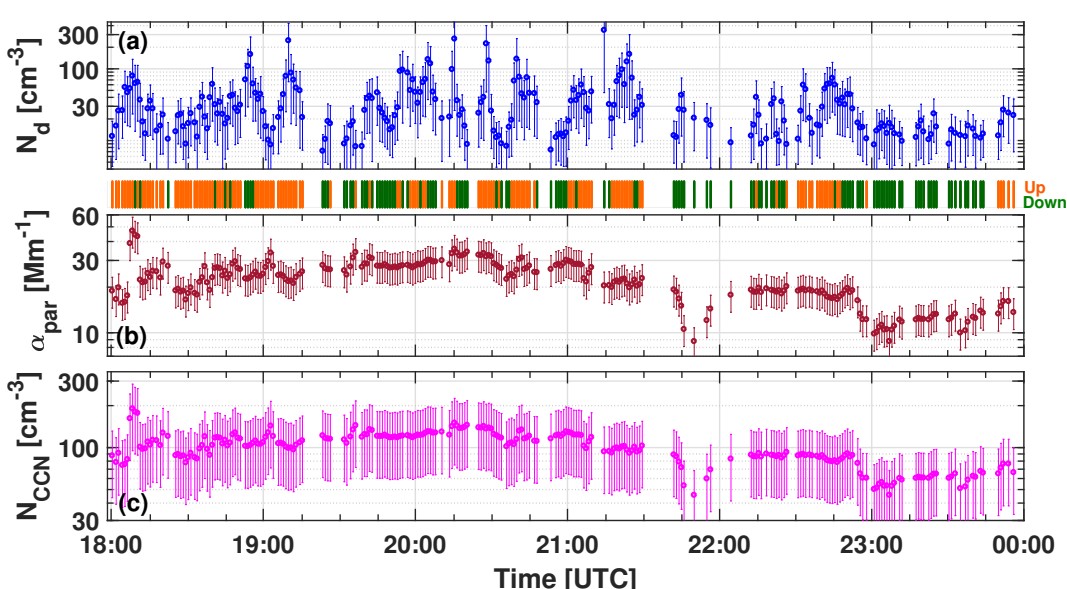

**Figure 9.** (a) Cloud droplet number concentration $N_\mathrm{d}$ for the height of $z_\mathrm{ref} = z_\mathrm{bot} + 75$ m within the stratocumulus layer shown in Fig. 8, (below a) vertical wind indicator (orange: updraft, green: downdraft), (b) particle extinction coefficient $\alpha_\mathrm{par}$ (mean value for the height range from 375 to 600 m below cloud base), and (c) CCN concentration $N_\mathrm{CCN}$ obtained from the extinction coefficient (in b) by using the marine conversion parameters (Sect. 6 in part 1) (Jimenez et al., 2020). Temporal resolution is one minute. Error bars indicate the uncertainty range.

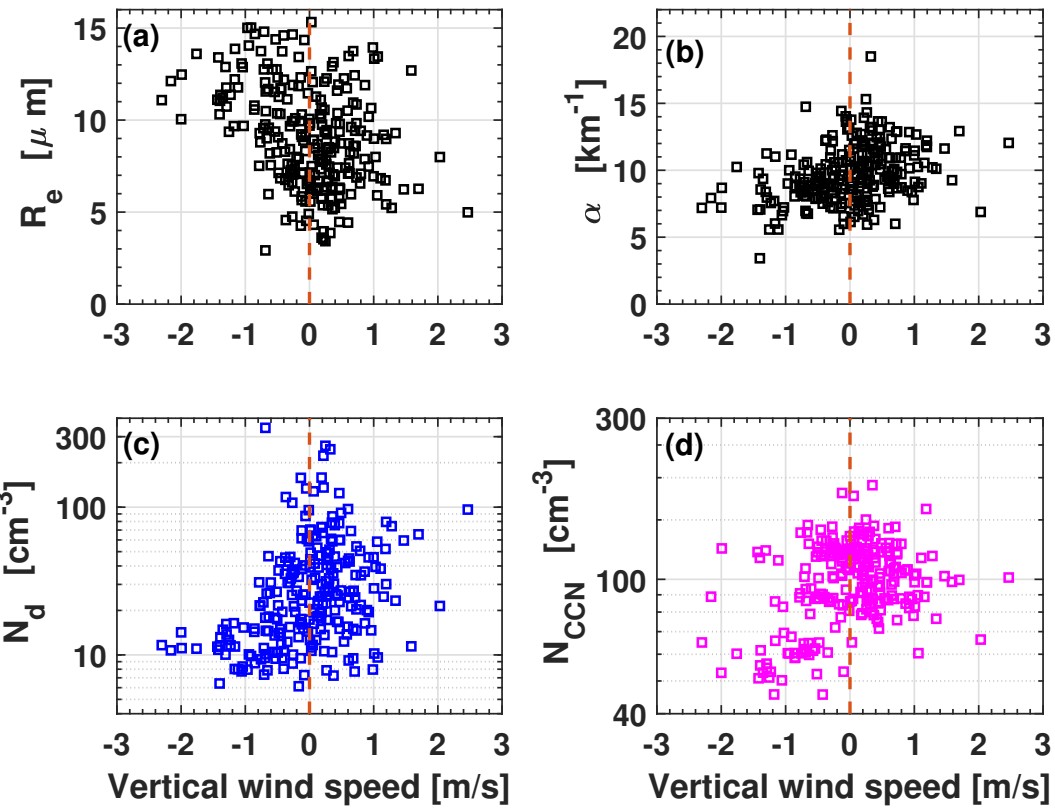

**Figure 10.** Correlation of retrieved cloud properties (droplet effective radius $R_e$, number concentration $N_d$, and 532 nm light-extinction coefficient $\alpha$) and aerosol CCN concentration $N_{CCN}$ (for a fixed water supersaturation level of 0.2%) versus vertical wind measured with Doppler lidar at cloud base. The data in Fig. 9 are used. The data are observed with one minute temporal resolution in a well-developed stratocumulus deck at the top of the pristine marine boundary layer.

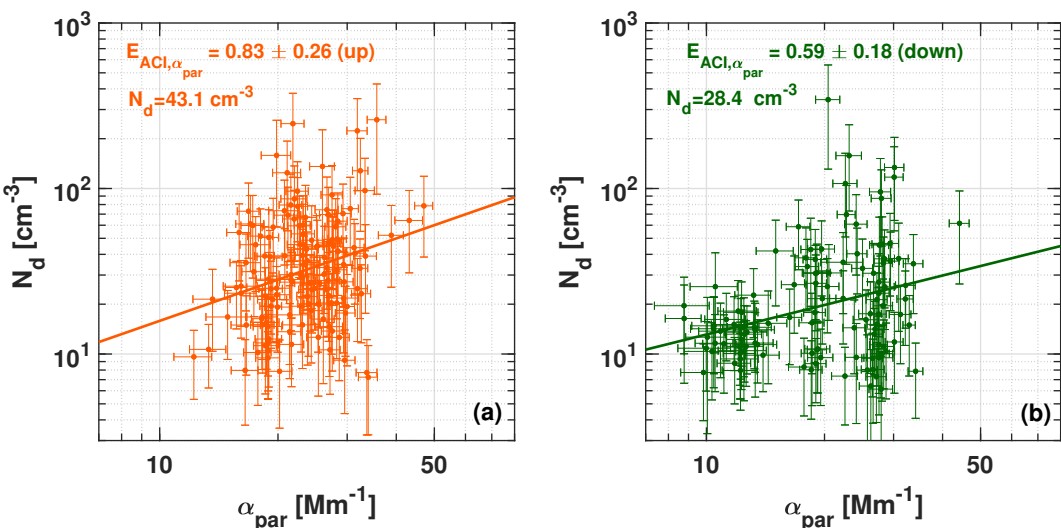

**Figure 11.** Cloud droplet number concentration $N_d$ (for height $z_{ref}$=75 m above cloud base) vs. aerosol particle extinction coefficient $\alpha_{par}$, separately for (a) updraft and (b) downdraft periods. The $N_d$ and $\alpha_{par}$ values shown in Fig. 9 are used (375-600m below cloud base). In total, almost 260 values were available for the regression analysis. Error bars show the uncertainties in the $N_d$ and $\alpha_{par}$ values. The linear regression fits a straight line to the $\log N_d$-$\log \alpha_{par}$ data field with the slope $E_{ACI,\alpha_{par}}$=0.87±0.26 (orange slope) and 0.58±0.17 (green slope). The mean droplet number concentration (given as numbers) was about 50% higher during updraft than during downdraft periods.



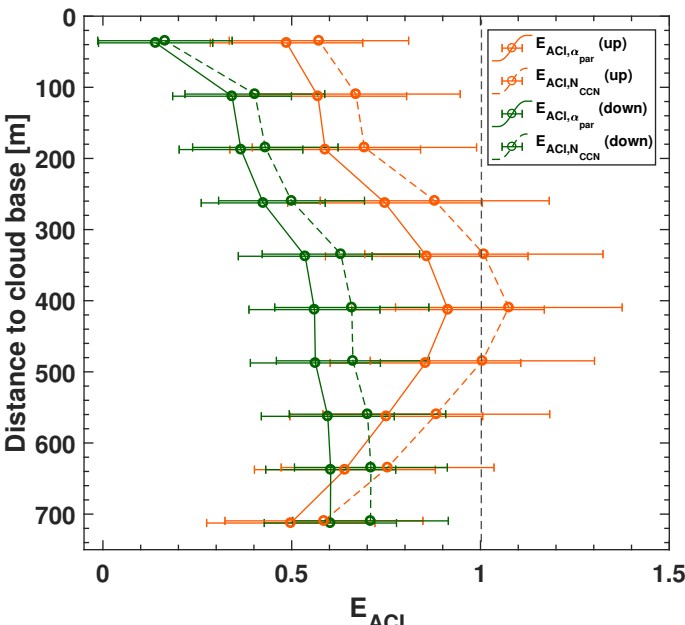

**Figure 12.** ACI efficiency parameter $E_{\mathrm{ACI}}$ (see Table 1 and Sect. 6 in part 1 for detailed explanations) as a function of $N_{\mathrm{d}}$ and $\alpha_{\mathrm{par}}$ (solid curves, $E_{\mathrm{ACI},\alpha_{\mathrm{par}}}$) and as function of $N_{\mathrm{d}}$ and $N_{\mathrm{CCN}}$ (dashed curve, $E_{\mathrm{ACI},N_{\mathrm{CCN}}}$), separated for updraft periods (orange) and downdraft periods (green). Different values of aerosol proxies $\alpha_{\mathrm{par}}$ and $N_{\mathrm{CCN}}$ for different layers (with 75 m vertical depth and increasing distance from cloud base towards lower heights) are considered in the calculations of the four $E_{\mathrm{ACI}}$ parameters (as explained in Fig.11). The ACI efficiencies are assigned to the center heights of these 75 m deep aerosol layers for which the aerosol proxies were determined. $E_{\mathrm{ACI}}$ values around 400 m below cloud base are obviously not affected by aerosol water-uptake effects which tend to widely smooth out a well defined and strong correlation between aerosol proxy and cloud droplet number concentration. Error bars indicate the uncertainty in the determination of the slopes of the linear regression analysis.