# Peer review of "The dual-field-of-view polarization lidar technique: A new concept in monitoring aerosol effects in liquid-water clouds — Case studies"

_Atmospheric Chemistry and Physics, 2020_

## Referee Comment (RC1) · Anonymous Referee #1 · 1 Aug 2020

Authors present a newly developed technique, based on use of dual FOV depolarization lidar measurements, for estimation of the cloud droplets concentration and effective radius. I should say that this is very interesting and well written manuscript, which is definitely suitable for publishing in ACP. The modification of the lidar, to get DFOV capability, looks rather simple, so can be used by many lidar groups. The theoretical background of DFOV approach is given in the companion manuscript. I don't feel confident enough to judge, how solid is this approach for derivation the droplets parameters , but comparison of effective radii obtained with DFOV method and results obtained from radar reflectivity shows very impressive agreement. In their approach, authors use two FOV: 1 and 2 mrad. I wonder, how optimal is this choice. Probably, the larger

the difference between "in" and "out" FOV, the more accurate should be results. Is any potential for farther optimization of this method? I had difficulty also to understand the goal of Fig.10. What information does it contribute? And again, this is very impressive, high quality research
* * *

---

## Referee Comment (RC2) · Anonymous Referee #3 · 3 Aug 2020

In this manuscript, the authors present a novel technique that is based on a dual field of view depolarization lidar. This setup allows the calculation of multiple cloud properties near the cloud base and the technique can be applied without the need of long averaging. This facilitates the monitoring of aerosol-cloud interactions based solely on lidars. The technique is rather interesting with great potential for future applications and adaptations and the manuscript is well written. I recommend this paper for publication in the ACP.

Below some suggestions:

1) To me it is not clear why figure 10 is necessary for the analysis. Differences can be

[Figure]

seen in the cloud droplet concentration and the effective radius between updrafts and downdrafts. But this is a single case. Is this behavior expected or systematic? It would be interesting to see more cases and determine weather the observed correlations are systematic but this is probably out of the scope of this paper.

2) In Fig. 11 and 12, it can be seen that the aerosol cloud interaction is stronger for the updrafts than it is for the downdrafts. The authors could mention some physical processes that would explain this behavior.

---

## Referee Comment (RC3) · Anonymous Referee #2 · 4 Aug 2020

This manuscript brings new observational skills to the problem of quantifying aerosol-cloud interactions using surface-based instrumentation. It brings an impressive array of new technical developments to simultaneously quantify cloud microphysical properties, updraft velocities, and sub-cloud aerosols. The downside of this paper in this reviewer's mind is that (i) the uncertainties in the retrievals – this paper and others like it – are large enough that they are not useful constraints on the problem; and (ii) the view of the cloud system as a vertical column in which one can learn about microphysics is a flawed one. That being said, I don't have a problem with the paper being published after some major revisions are made. These should persuade the reader of the utility of the approach. I offer some thoughts below.

[Figure]

Major comments

1) The ground-based system looks up at the clouds, and has the advantage of measuring aerosols beneath the clouds rather than between the clouds as with space measurements. But there is a significant over-interpretation of the relationship between below-cloud aerosol, updraft and drop concentration because the cloud system is almost never static and advection/diffusion mixes different drop concentrations together. Thus, a metric like the activated fraction doesn't make sense. It is also no wonder that the "ACI metric" is so noisy to the point that one wonders how useful it really is. Even without the uncertainties in Nd and particle extinction, the scatter is so large (Figure 11). This is a general criticism of this approach. You need to articulate much better what it is good for?

2) Since this is an "example paper", you should clarify what the applications are. For various reasons it's not an approach that teaches us about cloud microphysics. The system sees some integrated view of all processes and trying to sort that all out in the presence of advection and entrainment is not a feasible approach for elucidating processes – especially given the large uncertainties.

It could be useful for constraining ACI in climate models, or as an adjunct to satellite studies that attempt to do so but with poorer accuracy. But then how well would you need to measure ACI so that it would be useful for this purpose. And you would need to consider the big difference in measurement/modeling scales. Quaas et al. (2009, ACP) addresses some of these issues by comparing ACI metrics at a ground site with ACI metrics in models. Frankly, I find this question of usefulness very challenging because there are so many uncertainties. If I had the answer I would be happy to share it.

3) For case 1, the speciation in the mixed phase cloud seems useful. My major concerns relate to the physical interpretation of the 2nd case. Section 4.2 rambles and provides no clarity. Numbers are reasonable but with the large uncertainties, what do we learn?

[Figure]

Examples: By means of the new dual-FOV polarization lidar technique, cloud and aerosol information can be derived with high temporal resolution which allows us to resolve different phases of cloud evolution and life cycle and to investigate the impact of individual updrafts on the droplet nucleation rate, droplet growth and corresponding evolution of the effective radius, and the Nd–NCCN relationship in very large detail. This cannot be true. Individual updrafts don't reflect on activation in the column. What the system measures overhead is the net result of upstream activation, mixing, condensation, collision-coalescence (perhaps), and all other processes that shape the size distribution on its way to the volume that you sample. Updrafts vs. Downdrafts. If you want to sort by updrafts vs. downdrafts, you would need to take into account the typical size of large eddies. The 'instantaneous' ($\sim$ 60s) sample likely measures the effects of upstream updrafts and downdrafts given the size of typical eddies (Fig. 8). A weak decrease of the cloud extinction coefficient is visible when going from updraft to downdraft conditions in line with the dissolution of droplets. (I assume you mean evaporation of droplets.) One can't make such a statement because evaporating drops will be larger than growing drops, all else equal, because of asymmetries associated with solute and curvature terms (Korolev 1995, JAS). So perhaps all else is not equal. But again what do we learn? New droplet formation and growth of existing droplets by water uptake led to a slight increase of the cloud extinction coefficient in many cases. A reasonable but not at all useful statement. A weak reduction of the mean effective radius during upward motions may indicate new droplet nucleation in the presence of existing droplets. Again, reasonable but not at all enlightening given the uncertainty in drop effective radius. Maybe it indicates lower liquid water content, which could come from a variety of processes or from uncertainty in the measurement. 4) Regarding uncertainties: The lowest part of Sc clouds (which case 2 appears to be) is adiabatic. Why do you assume sub-adiabatic? How much influence does this have? How big of an error does the extinction/backscatter ratio have on the sub-cloud CCN?

You mention 500 m below cloud base as being "dry". Is this really so? Hysteresis in the efflorescence curve will hold on to water vapor down to small RH, especially in the

presence of organics.

The choice of CCN proxy has a very large effect on the metrics. What value is the right one? (E.g., Shinozuka 2015). I don't believe there is an answer to this question. And because of this I don't see a way to use this approach for quantification.

5) I appreciated the clarity in Fig. 5 showing source of uncertainty in Re. Fig. 6 left me confused. One of the Frisch approaches is to calculate Re from Z assuming a fixed Nd. In such a case you can tune Nd to get the Re over the range that you want. So the statement "Good agreement was found" is misleading when it comes to the comparison with radar and fixed N. Radar is so sensitive to large drops and I would put more faith into optical measurements. Did you look at deriving mean Re from cloud optical depth and liquid water path? It's a simple, straightforward approach to retrieving the Re from exactly the moments of interest (2nd and 3rd). It's unpretentious and quite solid if the uncertainties are known.

Other comments

6) Please provide historical context to the work that has been done trying to quantify ACI from the surface. The paper should quote early and usually similarly flawed work by various authors including Kim, Schwartz et al. (2003, JGR), Feingold et al. (2003, GRL), Garrett et al. (2004, GRL), Sena et al. (2016, ACP) to name some that come to mind. Some used optical depth and microwave radiometer to measure layer-mean cloud drop size; others used radar and microwave radiometer. Some used surface aerosol measurements, and others lidar profiles. Please give a little recap of the advantages and disadvantages of these older approaches.

7) I was frustrated by references to figures and equations in Part 1. E.g., table 1 references equations in Part 1! You shouldn't expect the reader to have all of those details, or to have to read two papers in parallel.

---

## Author Comment (AC1) · 6 Oct 2020

**Author response to RC1, RC2 and RC3 (acp-2020-489)**

**Dear Editor,**

We thank the reviewers for careful reading and constructive suggestions and criticism. In this document we would like provide our answers to the three Reviews.

Our answers are marked in bold text. In the revised version of the manuscript all changes due to the expressed suggestions were highlighted using different colors. Blue for changes due to Referee comment RC1, green for changes due to RC2, and red for changes due to RC3.

**RC1**

**Anonymous Referee #1**

Authors present a newly developed technique, based on use of dual FOV depolarization lidar measurements, for estimation of the cloud droplets concentration and effective radius. I should say that this is very interesting and well written manuscript, which is definitely suitable for publishing in ACP. The modification of the lidar, to get DFOV capability, looks rather simple, so can be used by many lidar groups. The theoretical background of DFOV approach is given in the companion manuscript. I don't feel confident enough to judge, how solid is this approach for derivation the droplets parameters, but comparison of effective radii obtained with DFOV method and results obtained from radar reflectivity shows very impressive agreement.

1) In their approach, authors use two FOV: 1 and 2 mrad. I wonder, how optimal is this choice? Probably, the larger the difference between "in" and "out" FOV, the more accurate should be results. Is any potential for farther optimization of this method?

Yes, according to the simulations (figure below), the larger the contrast between the two FOVs, the more sensitive the system and the more accurate should thus be the retrieval. We calculated these sensitivities for several pairs of FOVs and cloud-base height, but at the end we excluded the figure below from the manuscript as it already contains a lot of information.

So, the main result is simple (figure, right panel) and is written in the manuscript. On the other hand, and this is also explained in the manuscript, in case of strong (horizontal and vertical) cloud inhomogeneity the two observations with two different FOVs may lead to quite different results as a function of this inhomogeneity because different cloud volumes are observed with the two different FOVs. In our case, and as we wrote in the manuscript, we checked the potential impact of horizontal inhomogeneities by correlating the separately measured values of depolarization ratios for FOV-in and FOV-out. Quoting to the manuscript: the scatter in the data (see the next figure below, not included in the manuscript) was very low and did not indicate any significant influence of horizontal cloud inhomogeneities on the measured parameters from which the effective radius and extinction coefficient of the cloud are retrieved. Existing inhomogeneity were likely smoothed out when averaging over time. We see a confirmation of this in the coherent structures in the time series of the retrieved cloud products.

The choice of the pair of FOVs responds to a compromise between these two features. Furthermore, the selection of 1 and 2 mrad is guided by the fact that the used Polly instrument already has a polarization channel at FOV=1.0 mrad and one elastic channel at 2.0 mrad. This makes the further upgrade towards a dual FOV polarization system straight forward and easy to realize, as written in the manuscript.

We removed the word 'optimum' from the text.

2) I had difficulty also to understand the goal of Fig.10. What information does it contribute?

Case study 2 (Figs. 8-12) is presented to show the full capability of a dual FOV lidar (in combination with a Doppler lidar) to contribute to aerosol and cloud research. If we have continuous observations with 60s resolution (here over 6 hours), then the next logical and consequent step are correlation studies between the different aerosol, cloud, and vertical wind parameters. And when doing correlations the first approach is to analyze: To what extent do vertical motions influence the observed cloud products,

and this is shown in Fig.10. This was the basic motivation for Fig.10. The other motivation or goal was to highlight that we are able to see cloud evolution in terms of aerosol, cloud, and up/downdraft properties with high temporal resolution.

We expected a quite clear relationship between updraft and droplet nucleation (and droplet effective radius). And we are sure that many scientist believe that this should be the case. Some experts, and one of these experts is reviewer RC3 (see the last part of the reply letter) do not expect that because of the complexity of cloud processes, features, and characteristics. Triggered by your comments and the ones of RC3 we changed and (hopefully) improved the discussion of Fig 10, and provide the reasons for the found correlation features found in Fig.10 in more detail (please see the revised version of Section 4.2).

And again, this is very impressive, high quality research.

Thank you!

**RC2**

**Anonymous Referee #3**

**Received and published: 3 August 2020**

In this manuscript, the authors present a novel technique that is based on a dual field of view depolarization lidar. This setup allows the calculation of multiple cloud properties near the cloud base and the technique can be applied without the need of long averaging. This facilitates the monitoring of aerosol-cloud interactions based solely on lidars. The technique is rather interesting with great potential for future applications and adaptations and the manuscript is well written. I recommend this paper for publication in the ACP.

Below some suggestions:

1) To me it is not clear why figure 10 is necessary for the analysis. Differences can be seen in the cloud droplet concentration and the effective radius between updrafts and downdrafts. But this is a single case. Is this behavior expected or systematic? It would be interesting to see more cases and determine whether the observed correlations are systematic but this is probably out of the scope of this paper.

As mentioned above (RC1 reply, point 2), the case study 2 (Figs. 8-12) is presented to show the full potential of a dual FOV lidar (in combination with a Doppler lidar) to contribute to aerosol and cloud research. And if we have continuous observations with 60s resolution (and this over many hours), then the next logical and consequent step are correlation studies between the different aerosol, cloud, and vertical wind parameters. And when doing correlations the first approach is to analyze: To what extent do vertical motions influence the observed cloud products, and this is shown in Fig.10.

You are right, this is just a single case. But at the moment we deal with two manuscripts (part 1 and part 2) that introduce a new method. In follow-up papers we will show more examples and case studies. We are presently working on a contrasting study of aerosol-cloud interaction, performed with a dual FOV lidar at Dushanbe, Tajikistan (polluted and dust Central Asia) and in Punta Arenas (clean marine conditions) during two long-term field campaigns. All the already investigated case studies show (more or less) the same scatter as shown in Fig.10.

Triggered by your comment (and the ones from RC1 and RC3) we improved at least the discussion of Fig. 10 (see revised version of Section 4.2) but also say that this just ONE case....

2) In Fig. 11 and 12, it can be seen that the aerosol cloud interaction is stronger for the updrafts than it is for the downdrafts. The authors could mention some physical processes that would explain this behavior.

Again, motivated by your comment, but also forced by the critical comments of reviewer RC3 we improved (and enlarged) the discussions of the entire case study (Figs.8-12), and give the reasons (physical processes that explain the found behavior) more clearly now (see revised section 4.2). As you will see, there are many processes and interactions that prohibit a (more) clear difference in the findings for downdraft and updraft periods. All in all, we now emphasizes in the manuscript (even more clearly) that we need to be cautious when interpreting our results because, besides droplet nucleation, other processes such as entrainment, collision/coalescence or evaporation of droplets play a role.

**RC3**

**Anonymous Referee #2**

This manuscript brings new observational skills to the problem of quantifying aerosol cloud interactions using surface-based instrumentation. It brings an impressive array of new technical developments to simultaneously quantify cloud microphysical properties, updraft velocities, and sub-cloud aerosols. The downside of this paper in this reviewer's mind is that (i) the uncertainties in the retrievals – this paper and others like it – are large enough that they are not useful constraints on the problem; and (ii) the view of the cloud system as a vertical column in which one can learn about microphysics is a flawed one. That being said, I don't have a problem with the paper being published after some major revisions are made. These should persuade the reader of the utility of the approach. I offer some thoughts below.

**With respect to the 'downside' arguments the glass seems to be more than half empty. But we are convinced that the glass is more than half full!**

Before we start the reply, we should clearly say: We are very happy to have this clear, deep, and openminded review provided by an aerosol-cloud interaction expert. Thank you for all the critical statements. And we noticed that YOU spent considerable time to write this review and even to provide summarizing concluding remarks that helped us a lot to see our own work in a more critical way.

Let us begin with the large uncertainties. They are a result of a traditional error propagation analysis. Such an approach always leads to large uncertainty ranges. But typical uncertainties (determined in field experiments with opportunities to compare our retrieval products with respective in situ measurements) were found to be much smaller. Furthermore, even if the uncertainties are large, the harmonic temporal evolutions and coherent variability in all the observed time series tell us a lot about aerosol-cloud interaction at given meteorological conditions. The found variations are trustworthy. This is, at least, what we see from our numerous data analysis efforts we did during the last months (Punta Arenas, Dushanbe, and even now North Pole Polarstern Polly data).

And to the second point (the view of the cloud system.... is a flawed one). Yes that may be true, but what does it help? Our basic message and motivation is always that we need to make progress towards continuous monitoring of cloud properties at given aerosol and meteorological conditions..... and all this at heights where the clouds develop. And our 'lidar-biased' opinion is that this is only possible with active remote sensing, that means with profiling techniques (and not by means of passive remote sensing and all these column-integrated approaches...).

Sure, our proposed dual FOV lidar method is far from being perfect (we have just information of microphysical properties for the cloud base region), but future developments in lidar, radar, and all these CLOUDNET-like instrumental setups and corresponding developments in data combination efforts will certainly improve the situation ...and the dual FOV lidar may then play an important role in the upcoming years... providing a constant flow of liquid water cloud properties. And just to continue and finalize this vision. The most promising way is then to combine all these CLOUDNET observations (obtained within continental scale networks) with cloud resolving model work.... to improve the models and the knowledge about aerosol-cloud interaction... (and to reduce in this way the 'flawed' situation). Sorry, for this long text without making progress with the reply...

**Major comments**

1) The ground-based system looks up at the clouds, and has the advantage of measuring aerosols beneath the clouds rather than between the clouds as with space measurements.

But there is a significant over-interpretation of the relationship between below-cloud aerosol, updraft and drop concentration because the cloud system is almost never static and advection/diffusion mixes different drop concentrations together. Thus, a metric like the activated fraction doesn't make sense. It is also no wonder that the "ACI metric" is so noisy to the point that one wonders how useful it really is. Even without the uncertainties in Nd and particle extinction, the scatter is so large (Figure 11). This is a general criticism of this approach. You need to articulate much better what it is good for?

We used 'activation fraction' because we found it in other papers (referenced in the manuscript) and found good agreement with these findings. But you are right. Activation ratio is always ONE.... and our CCNC estimate expresses the CCNC for water super saturation of 0.2% and the true supersaturation is obviously lower, 0.1-0.15% so that less aerosol particles as indicated by CCNC became activated.

We removed 'activation ratio' from the text.

Regarding the scatter in the data. As you know as an expert in this field, this is nature (and not the uncertainties). There is no paper dealing with ACI based on aerosol and cloud data (in situ measured or with active remote sensing) in which this scatter is not present and shown.

But surprisingly, on the long term (and even in some lucky, single case studies as presented in our paper) ... the link between aerosol concentration and cloud microphysical properties is still visible. And the ACI values are in the range where they should be ... close to 1.

So now: What is our goal? We want to provide data on aerosols and clouds and vertical velocity with quite high temporal resolution, at cloud height level...., or near cloud base level. Furthermore, in future we want to be able to illuminate cloud processes (even for individual cloud life cycles), we want to see

links between aerosol levels and cloud microphysical properties, we want to support cloud-resolved modeling. We are not able to step forward from this point of cloud studies. We do not think about the radiative impact of liquid water clouds as a function of aerosol concentration (the classical Twomey effect). The method is unable to contribute anything in this direction.

And having this in mind, we organized and designed the paper with the two case studies. The first case study explains the methods and discusses uncertainties. And the second illuminates the potential of combined aerosol lidar, dual FOV lidar, and Doppler lidar.

And regarding case 2, we see the scatter in the observations as a clear results of natural processes and not as a result of the lidar uncertainties. And, based on all the data, we still get the impact of observed aerosol concentration levels on cloud properties..., this is our goal...to show that. As exercise, to smooth out the effect of small variations due to other processes we performed the retrieval of case 2 (23 Feb) at 20 min. resolution (not at 60s as in this manuscript). Here the tendency, with similar ACI index, can be clearly seen and the scatter is lower.

Our goal and contribution to ACI studies was already mentioned in our other papers (Schmidt et al., 2014, 2015). So, we will not repeat all this here.

We see a coherent tendency in the ACI (of all the data analyzed during the last months) and we believe that by considering more cases with similar conditions, a more robust and useful estimate of this ACI index may be possible. To have a first look into the ACI values for the full campaign period (more than one year at Punta Arenas), we show the figure below (monthly mean values of CDNC and CCNC are shown). This will be material for future articles.

**We improved the Introduction section (in red) and mention better our goals....**

2) Since this is an "example paper", you should clarify what the applications are. For various reasons it's not an approach that teaches us about cloud microphysics. The system sees some integrated view of all processes and trying to sort that all out in the presence of advection and entrainment is not a feasible approach for elucidating processes – especially given the large uncertainties. It could be useful for constraining ACI in climate models, or as an adjunct to satellite studies that attempt to do so but with poorer accuracy. But then how well would you need to measure ACI so that it would be useful for this purpose. And you would need to consider the big difference in measurement/modeling scales. Quaas et al. (2009, ACP) addresses some of these issues by comparing ACI metrics at a ground site with ACI metrics in models. Frankly, I find this question of usefulness very challenging because there are so many uncertainties. If I had the answer I would be happy to share it.

Maybe we did not get the point of the reviewer! So, we observe the atmosphere, and find complex structures and relationships. This is the first goal: We want to study and learn more about cloud evolution within given aerosol structures and at given vertical wind conditions. We do not take care too much of scales (sorry) and existing model approaches, we just try to analyze our data to get the products with the highest temporal resolution possible.

There is a clear gap in continuous monitoring, profiling and simultaneous characterization of aerosol and clouds and up/downdraft conditions. And then, in the next step, we try to find relationships and links between the different observables. Since we know that an aerosol effect on cloud properties should become visible during updraft events, we investigate that in detail. And then we compute the relationship between CCNC and CDNC and find reasonable ACI values..... This is what we do! This is the field of applications...

Ok, in this paper we have to clearly state that this is just ONE case. We must avoid to make 'general' statements. We checked the text for this and changed the text where it was necessary.

3) For case 1, the speciation in the mixed phase cloud seems useful. My major concerns relate to the physical interpretation of the 2nd case. Section 4.2 rambles and provides no clarity. Numbers are reasonable but with the large uncertainties, what do we learn?

We must say that we were never confused by the 'theoretically' high uncertainties. As with this reviewer, we find that the results are reasonable, and we find the consistency between our results and other previous independent studies important. Thus, we stepped forward including Figs. 10-12 in the manuscript.

The uncertainty analysis, as given here, is the typical approach in the lidar business, and to say it again, the typical uncertainties are much lower than the theoretical ones....

Examples: By means of the new dual-FOV polarization lidar technique, cloud and aerosol information can be derived with high temporal resolution which allows us to resolve different phases of cloud evolution and life cycle and to investigate the impact of individual updrafts on the droplet nucleation rate, droplet growth and corresponding evolution of the effective radius, and the Nd–NCCN relationship in very large detail. This cannot be true. Individual updrafts don't reflect on activation in the column. What the system measures overhead is the net result of upstream activation, mixing, condensation, collision-coalescence (perhaps), and all other processes that shape the size distribution on its way to the volume that you sample. Updrafts vs. Downdrafts. If you want to sort by updrafts vs. downdrafts, you would need to take into account the typical size of large eddies.

That may be true that individual updrafts don't reflect on activation in the column. But we know that droplet nucleation preferably takes place during updraft periods! And in our case study, the updraft periods were relatively large and mostly well defined, and most of the updraft speed values were between 10 and 70 cm/s, so in a good, calm, and reasonable range to study the link between updraft occurrence and speed, supersaturation, and cloud droplet nucleation. But we rephrased the text according to critical remarks of the reviewer (we used the stated arguments in the review) and thus indicate that the processes, impacts, and features are rather complex and simple conclusions cannot be drawn... (see the red parts in section 4.2).

The 'instantaneous' (\_ 60s) sample likely measures the effects of upstream updrafts and downdrafts given the size of typical eddies (Fig. 8). A weak decrease of the cloud extinction coefficient is visible when going from updraft to downdraft conditions in line with the dissolution of droplets. (I assume you mean evaporation of droplets.) One can't make such a statement because evaporating drops will be larger than growing drops, all else equal, because of asymmetries associated with solute and curvature terms (Korolev 1995, JAS). So perhaps all else is not equal. But again what do we learn? New droplet formation and growth of existing droplets by water uptake led to a slight increase of the cloud extinction coefficient in many cases. A reasonable but not at all useful statement. A weak reduction of the mean effective radius during upward motions may indicate new droplet nucleation in the presence of existing droplets. Again, reasonable but not at all enlightening given the uncertainty in drop effective radius. Maybe it indicates lower liquid water content, which could come from a variety of processes or from uncertainty in the measurement.

Yes, we mean evaporation of droplets...

We got the point and changed the text accordingly, as we stated just one comment before. We agree that we have to be careful with such statements. We changed the text as mentioned above.

But at the end, we have our observations, here in this case in an aged stratocumulus deck..., and can at least check whether we find an aerosol effect or not. Sure, again we need to state, this is just an example, please wait for the entire Punta Arenas data analysis ... (almost two years of data).

We learn that the world is quite complex and simple relationships between aerosol and cloud microphysics are not given, even if we resolve up and downdraft periods (including strength and size of eddies).

4) Regarding uncertainties: The lowest part of Sc clouds (which case 2 appears to be) is adiabatic. Why do you assume sub-adiabatic? How much influence does this have?

Our assumption of a sub-adiabatic cloud was mainly done because of the operational advantage of having a linearly increasing water content into the cloud. Whether the cloud is adiabatic or sub-adiabatic, the linearity is still there, changing only the slope among this two systems. In our approach, we do not perform calculations about the sub-adiabatic (or adiabatic) lapse rate, instead we determine the slope of the liquid water content from our retrieval.

Our first goal with our DFOV-Depolarization approach is to quantify the aerosol cloud interaction index for updraft periods, to which an adiabatic cloud may describe the scene properly, and also for downdraft periods where entrainment of dry air masses may reduce the lapse rate producing the socalled sub-adiabatic conditions. Again, whether is adiabatic or sub-adiabatic does not affect our approach. We leave the sub-adiabatic system, because this one also comprise the adiabatic case, i.e. the adiabaticity factor=1 (according to Merk et al., 2016).

How big of an error does the extinction/backscatter ratio have on the sub-cloud CCN? You mention 500 m below cloud base as being "dry". Is this really so? Hysteresis in the efflorescence curve will hold on to water vapor down to small RH, especially in the presence of organics. The choice of CCN proxy has a very large effect on the metrics. What value is the right one? (E.g., Shinozuka 2015). I don't believe there is an answer to this question. And because of this I don't see a way to use this approach for quantification.

The error that the extinction/backscatter ratio have on the extinction and therefore in the CCN is small (about 10%). To obtain the extinction profiles we considered a constant extinction/backscattering ratio (a.k.a. Lidar ratio) of 25 [sr] as typical for the marine boundary layer. This lidar ratio could not be determined during the measurement period, but it can be obtained during nighttime and during cloud-free periods, which was the case the day after (24 Feb from 02:00 to 07:00 UTC), so we determined the lidar ratio profiles in the boundary layer and the values varies from about 22 to 30 [sr] which corroborates that our assumption is appropriate.

Regarding the CCN estimates, we do not know whether the aerosol particles were completely dry. However, the ACI values were large and indicated at least almost dry aerosol conditions. Otherwise such high ACI values are impossible.

Shinosuka's (2015) approach is the only reasonable way to translate singe-wavelength lidar data into CCNC numbers. Feel free to accept it or not. However, we match the range of CCNC for the Southern Ocean. So, why should we skip that? And if we are able to measure backscatter and extinction profiles, and know that there is a link between optical properties and CCNC, why should we not use that? By the way, if you do not trust our CCNC numbers then you should especially not trust cloud droplet and CCNC retrievals from spaceborne remote sensing. And there are many papers on this.

5) I appreciated the clarity in Fig. 5 showing source of uncertainty in Re. Fig. 6 left me confused. One of the Frisch approaches is to calculate Re from Z assuming a fixed Nd. In such a case you can tune Nd to get the Re over the range that you want. So the statement "Good agreement was found" is misleading when it comes to the comparison with radar and fixed N. Radar is so sensitive to large drops and I would put more faith into optical measurements. Did you look at deriving mean Re from cloud optical depth and liquid water path? It's a simple, straightforward approach to retrieving the Re from exactly the moments of interest (2nd and 3rd). It's unpretentious and quite solid if the uncertainties are known.

We aimed to use the effective radius scaled to the LWP from the MWR, but the multiple cloud layer, liquid and ice, above and below the cloud hampered the application of this approach. Since the MWR can get radiation mostly from the first liquid layer, when redistributing the water content in all layers we got a strong reduction in the droplet size. To perform a simple retrieval we assumed a number concentration of 100  $cm^{-3}$ . It is true, that in this way one can tune this parameter until one gets in the range wanted. But given that in the Frisch approach the effective radius is proportional to the power of (1/6) of the number concentration, this tuning turns out small. If we assume a concentration of 50  $cm^{-3}$ , the effective radius increase 12%, and when we assume a concentration of 200  $cm^{-3}$ , the effective radius decrease about 11%. These a priori variations were considered in the error calculation.

**We did not change the text...**

**Other comments**

6) Please provide historical context to the work that has been done trying to quantify ACI from the surface. The paper should quote early and usually similarly flawed work by various authors including Kim, Schwartz et al. (2003, JGR), Feingold et al. (2003, GRL), Garrett et al. (2004, GRL), Sena et al. (2016, ACP) to name some that come to mind. Some used optical depth and microwave radiom

---

## Author Comment (AC3) · 6 Oct 2020

Dear Editor, Dear Reviewers,

Many thanks for your time and efforts.

Please find our reply letter in the supplement,

best wishes,

Cristofer Jimenez (on behalf of all authors)

Please also note the supplement to this comment:

[Figure]

https://acp.copernicus.org/preprints/acp-2020-489/acp-2020-489-AC3-supplement.pdf